# FHA-KITCHENS: A NOVEL DATASET FOR FINE-GRAINED HAND ACTION RECOGNITION IN KITCHEN SCENES

## ABSTRACT

A typical task in the field of video understanding is hand action recognition, which has a wide range of applications. Existing works either mainly focus on full-body actions, or the defined action categories are relatively coarse-grained. In this paper, we propose **FHA-Kitchens**, a novel dataset of fine-grained hand actions in kitchen scenes. In particular, we focus on human hand interaction regions and perform deep excavation to further refine hand action information and interaction regions. Our FHA-Kitchens dataset consists of 2,377 video clips and 30,047 images collected from 8 different types of dishes, and all hand interaction regions in each image are labeled with high-quality fine-grained action classes and bounding boxes. We represent the action information in each hand interaction region as a triplet, resulting in a total of 878 action triplets. Based on the constructed dataset, we benchmark representative action recognition and detection models on the following three tracks: (1) supervised learning for hand interaction region and object detection, (2) supervised learning for fine-grained hand action recognition, and (3) intra- and inter-class domain generalization for hand interaction region detection. The experimental results offer compelling empirical evidence that highlights the challenges inherent in fine-grained hand action recognition, while also shedding light on potential avenues for future research, particularly in relation to pre-training strategy, model design, and domain generalization. The dataset will be released at project website.

## 1 INTRODUCTION

Action recognition, a prominent task within the domain of video understanding, has garnered considerable attention and possesses broad applications across various fields Zhang & Tao (2020), including human computer interaction (HCI) Hu et al. (2022), smart homes Alaa et al. (2017), healthcare Ye et al. (2022), and the design and control of robot hands Palli et al. (2013). While there has been extensive research on action recognition concerning large-scale benchmarks Soomro et al. (2012); Carreira & Zisserman (2017) and advanced algorithms Wang et al. (2016); Feichtenhofer et al. (2019); Liu et al. (2022), relatively fewer studies have focused on the recognition of fine grained hand actions. This is primarily due to the extensive diversity of hand actions, the complex interaction situations, and the required fine-grained categorization of such actions, all of which pose significant challenges in terms of data collection and annotation. Nevertheless, given that a substantial portion of human actions in daily life originates from hand actions, the recognition of fine-grained hand actions assumes critical importance in both research and practical applications. Therefore, it is desirable to establish a large-scale benchmark that encompasses diverse fine-grained hand actions, as it would serve as a solid foundation for further research in this field.

In an effort to address the need for hand action recognition, datasets like MPII Cooking Activities Rohrbach et al. (2012) and EPIC-KITCHENS Damen et al. (2018) have been developed. Although these datasets have made efforts to study fine-grained hand actions, they still have certain limitations, including insufficient excavation of hand action information, coarse-grained representations of interaction actions (*e.g.*, "*cut*" rather than finer-grained "*<knife, cut slice, carrot>*"), a lack of localization of hand interaction regions, and insufficient attention to the relationships between interacting objects. These limitations pose significant obstacles to the research endeavors aimed at tackling the inherent challenges associated with fine-grained hand action recognition tasks.

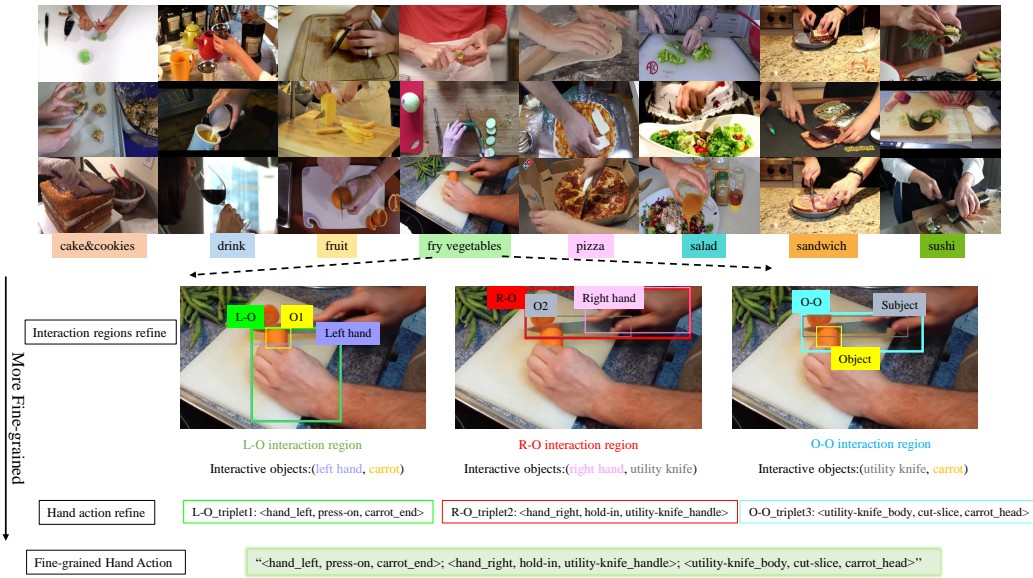

Figure 1: Overview of the **FHA-Kitchens** dataset. The top shows some frames extracted from 8 dish categories. The bottom illustrates the annotation process of fine-grained actions in *"fry vegetable"*.

In this paper, we present a novel dataset, namely the **FHA-Kitchens** dataset, which focuses on fine-grained hand actions observed in kitchen scenes. The FHA-Kitchens dataset encompasses a total of 2,377 video clips and 30,047 images, all extracted from eight different dish types (Figure 1 top). Each frame within the dataset is accompanied by meticulously labeled hand action information, featuring high-quality annotations of fine-grained action categories and bounding boxes. To create the FHA-Kitchens dataset, we derived data from publicly available large-scale action datasets Smaira et al. (2020), specifically targeting videos that were highly relevant to hand actions, and conducted frame extraction and cleaning processes. Subsequently, we engaged the expertise of ten voluntary annotators to meticulously label the interaction information of each hand. Notably, we divided the hand interaction regions into three distinct sub-regions: the **left hand-object** interaction region, the **right hand-object** interaction region, and the **object-object** interaction region. For each sub-interaction region, we provided bounding box annotations. Furthermore, we categorized hand actions into three distinct types to adequately capture the actions within the sub-interaction regions. Each sub-interaction region action was annotated using a triplet format, denoted as *<subject, action verb, object>*. Additionally, we took into account the "active-passive" relationships between interaction object pairs and the specific contact areas involved in the interaction actions. Consequently, our annotation process encompassed a total of nine dimensions (Figure 1 bottom), resulting in 878 annotated action triplets for hand actions. Finally, we organized the video frames based on the action triplet classes, ultimately generating 2,377 clips that represent distinct hand action triplet classes.

The FHA-Kitchens dataset provides valuable opportunities for advancing fine-grained hand action research, owing to its extensive diversity and high-dimensional representations of fine-grained hand actions. In light of this, we propose three distinct tracks as benchmarks for assessing representative action recognition and detection models. These tracks encompass: (1) supervised learning for hand interaction region and object detection (SL-D track), (2) supervised learning for fine-grained hand action recognition (SL-AR track), and (3) intra- and inter-class domain generalization for hand interaction region detection (DG track). In the SL-D track, we investigate the three representative methods, *i.e.*, Faster-RCNN Ren et al. (2015), YOLOX Ge et al. (2021), and Deformable DETR Zhu et al. (2020), with different backbones. In the SL-AR track, we train and evaluate several representative action recognition models (*e.g.*, TSN Wang et al. (2016), SlowFast Feichtenhofer et al. (2019), VideoSwin Liu et al. (2022), etc.) and investigate the influence of parameter initialization. In the DG track, we investigate the generalization ability of detection models regarding intra-class generalization, where the model is trained on specific sub-category instances within the same parent categories and subsequently tested on other sub-categories within the same parent categories, and inter-class generalization, where the model is trained on all instances encompassing a specific parent category and then tested on other different parent categories. The experimental findings furnish compelling

empirical evidence that provides valuable insights into the challenges associated with fine-grained hand action recognition and shed light on potential avenues for future research, particularly in relation to pre-training strategy, model design, and domain generalization.

The main contributions of this paper are three-folds: (1) We introduce FHA-Kitchens, a novel dataset for fine-grained hand action recognition, encompassing 2,377 video clips and 30,047 images with high-quality annotations; (2) We propose to employ triplets to represent action in each hand sub-interaction region while also considering the "active-passive" relationships between interacting objects and their contact areas, resulting in 878 action triplets covering 131 action verbs and 384 object nouns; (3) Based on FHA-Kitchens, we study several challenging yet unexplored questions in this field by benchmarking representative action recognition and detection models on the SL-D, SL-AR, and DG tracks. The obtained compelling empirical evidence highlights the challenges inherent in fine-grained hand action recognition, while also illuminating avenues for future research.

## 2 RELATED WORK

### 2.1 ACTION RECOGNITION DATASETS

Existing studies on action recognition datasets can be divided into two main categories based on the types of actions: full-body action and part-body action. Pioneering action recognition datasets, such as KTH Schuldt et al. (2004) and Weizmann Blank et al. (2005), have played a pivotal role in the advancement of this field, inspiring subsequent endeavors in constructing more challenging datasets, such as UCF101 Soomro et al. (2012), Kinetics Carreira & Zisserman (2017); Carreira et al. (2018; 2019), ActivityNet Heilbron et al. (2015), FineGym Shao et al. (2020), and others Monfort et al. (2019); Heilbron et al. (2018); Jhuang et al. (2013); Sigurdsson et al. (2016); Zhao et al. (2019). While these datasets primarily focus on full-body actions, lacking fine-grained action information from specific body parts. Datasets like MPII Cooking Activities Rohrbach et al. (2012) and EPIC-KITCHENS Damen et al. (2018) fill this gap. They refine the action verb part and consider interaction objects, but they fail to address the localization of interaction regions or the relationships between interacting objects. Furthermore, they represent hand action information only based on the appearance of the hand action pose. However, due to the complexity and diversity of hand actions, it is insufficient to represent hand action information only based on the appearance of the hand-object interaction. To mitigate this issue, our FHA-Kitchens dataset sets itself apart from existing datasets in **three key aspects:** (1) *Action Interaction Regions*: We meticulously annotate hand interaction regions and their corresponding objects using bounding boxes; (2) *Action Representation*: We categorize hand actions into three classes based on sub-interaction regions and employ a triplet to express each sub-interaction region action, thereby expanding the dimensionality to 9; and (3) *Interacting Objects*: In contrast to previous datasets that solely considered the active force provider, we focus on both the active and passive relationships between interacting objects and capture their contact areas.

### 2.2 ACTION DETECTION DATASETS

Compared to action recognition datasets, there are fewer datasets available for action detection Gu et al. (2018); Li et al. (2020). This is due to the intricate and diverse nature of hand actions, as well as the relatively smaller size of interacting objects, which introduce challenges such as occlusion and truncation. The AVA dataset Gu et al. (2018) focuses on human action localization, providing bounding box annotations for each person. However, this dataset provides the action verbs that are coarse-grained (*e.g.*, "*sit*", "*write*", and "*stand*") and does not account for the specific interacting objects involved in the actions and their relationships. In our dataset, we surpass these limitations by providing precise bounding box annotations for each hand sub-interaction region. Moreover, we refine the expression of action verbs and incorporate information about the interacting objects within each interaction region, thereby enhancing the granularity and contextual information of hand actions. A comprehensive comparison between FHA-kitchens and existing datasets is presented in Table 1.

### 2.3 ACTION RECOGNITION METHODS

On the one hand, existing action recognition methods can be categorized into coarse-grained Dalal & Triggs (2005); Dalal et al. (2006) and fine-grained Ni et al. (2016); Munro & Damen (2020);

Table 1: Comparison of relevant datasets. AR: Action Recognition. ARL: Action Region Localization. HIRD: Hand Interaction Region Detection. OD: Object Detection. ACat.: Action Category. OCat.: Object Category. Dim: Action Dimension. Box: Box Annotation of Action Region.

| Dataset | Year | Ego | #Clip | Ave.Len | #Frame | #ACat. | #Verb | #OCat. | Dim | Box | Task |
|---|---|---|---|---|---|---|---|---|---|---|---|
| **Human full-body dataset** | | | | | | | | | | | |
| UCF101 Soomro et al. (2012) | 2012 | × | 13.3K | ~6s | - | 101 | - | - | 1 | × | AR |
| ActivityNet Heilbron et al. (2015) | 2015 | × | 28K | [5,10]m | - | 203 | - | - | 1 | × | AR |
| Kinetics400 Carreira & Zisserman (2017) | 2017 | × | 306K | 10s | - | 400 | 359 | 318 | 2 | × | AR |
| Kinetics600 Carreira et al. (2018) | 2018 | × | 496K | 10s | - | 600 | 550 | 502 | 2 | × | AR |
| Kinetics700 Carreira et al. (2019) | 2019 | × | 650K | 10s | - | 700 | 644 | 591 | 2 | × | AR |
| AVA Gu et al. (2018) | 2018 | × | 430 | 15m | - | 80 | 80 | 0 | 3 | ✓ | AR,ARL |
| AVA-kinetics Li et al. (2020) | 2020 | × | 230K | 15m,10s | - | 80 | 80 | 0 | 3 | ✓ | AR,ARL |
| FineGym Shao et al. (2020) | 2020 | × | 32K | 10m | - | 530 | 530 | 0 | 3 | × | AR |
| **Hand dataset** | | | | | | | | | | | |
| MPII cooking Rohrbach et al. (2012) | 2012 | × | 5,609 | 15m | 881K | 65 | 65 | 0 | 1 | × | AR |
| EPIC-KITCHENS Damen et al. (2018) | 2018 | ✓ | 39.6K | 3.7±5.6s | 11.5M | 149 | 125 | 323 | 2 | × | AR,OD |
| **FHA-Kitchens** | **2023** | **✓** | **2377** | **3m** | **30,047** | **878** | **131** | **384** | **9** | **✓** | **AR,ARL,HIRD,OD** |

Hong et al. (2021); Liu et al. (2020). These methods heavily rely on the specific dataset used and offer tailored solutions to the associated challenges. On the other hand, according to the model architecture, action recognition methods can also be broadly summarized into three groups. The first group employs a 2D CNN Simonyan & Zisserman (2014); Wang et al. (2018a); Donahue et al. (2015); Feichtenhofer et al. (2016) to learn frame-level semantics and then aggregate them temporally using 1D modules. For example, TSN Wang et al. (2016) divides an action instance into multiple segments, represents it with a sparse sampling scheme, and applies average pooling to fuse predictions from each frame. TRN Zhou et al. (2018) and TSM Lin et al. (2019) replace pooling with temporal reasoning and shift modules, respectively. The second group directly utilizes a 3D CNN Carreira & Zisserman (2017); Wang et al. (2018b); Feichtenhofer et al. (2019); Tran et al. (2018); Diba et al. (2017) to capture spatial-temporal semantics, such as I3D Carreira & Zisserman (2017) and SlowFast Feichtenhofer et al. (2019). The third group utilizes transformers for action recognition tasks, such as the recent methods VideoSwin Liu et al. (2022), VideoMAE V2 Wang et al. (2023), and Hiera Ryali et al. (2023). In addition to action recognition, other video understanding tasks have also garnered research attention, including action detection and localization Wu et al. (2019); Girdhar et al. (2019); Xu et al. (2017); Zhao et al. (2017), action segmentation Lea et al. (2017); Ding & Xu (2018), and action generation Li et al. (2018); Sun et al. (2019).

## 3 DATASET

This section introduces the FHA-Kitchens dataset (Figure 1). Specifically, we describe the data collection and annotation pipeline and present statistics regarding different aspects of the dataset.

### 3.1 DATA COLLECTION AND ORGANIZATION

**Data Collection.** Our dataset is derived from the large-scale action dataset Kinetics 700_2020 Smaira et al. (2020), which comprises approximately 650K YouTube video clips and over 700 action categories. However, as the Kinetics 700 dataset primarily focuses on person-level actions, most of the videos capture full-body actions rather than specific body parts. To narrow our focus to hand actions, we performed filtering and processing operations on the original videos, including the following three steps. (1) First, we observed that kitchen scenes often featured hand actions, with video content prominently showcasing human hand parts. Therefore, we sought out and extracted relevant videos that were set against a kitchen backdrop. (2)Then, to ensure the quality of the dataset, we selectively chose videos with higher resolutions. Specifically, 87% of the videos were recorded at 1,280 × 720 resolution, while another 13% had a shorter side of 480. Additionally, 67% of the videos were captured at 30 frames per second (fps), and another 33% were recorded at 24~25 fps. (3) Subsequently, we imposed a duration constraint on the videos, ranging from 30 seconds to 5 minutes, to exclude excessively long-duration videos. This constraint aimed to maintain a balanced distribution within the sample space. Finally, we collected a total of 30 videos, amounting to 84.22 minutes of footage, encompassing 8 distinct types of dishes.

**Data Organization.** The collected video data was reorganized and cleaned to align with our annotation criteria (Section 3.2). First, we split the collected video data into individual frames, as our

annotated units are frames. Subsequently, we conducted further cleaning of the frames by excluding those that did not depict hands or exhibited meaningless hand actions. This cleaning process took into consideration factors such as occlusion, frame quality (*i.e.*, without significant blur, subtitles, and logos), meaningful hand actions, and frame continuity. As a result, we obtained a total of 30,047 high-quality candidate video frames containing diverse hand actions for our FHA-Kitchens dataset. Compared to the initial collection, 113,436 frames were discarded during the cleaning process.

## 3.2 DATA ANNOTATION

To ensure high-quality annotation of hand actions for each frame, we recruited 10 voluntary annotators, whose responsibility was to annotate fine-grained action triplet classes and bounding boxes for each hand interaction region. In order to enhance annotation efficiency, we implemented a parallel annotation approach. The annotation of action triplets was carried out on the Amazon Mechanical Turk platform, while the bounding box annotation was facilitated using the LabelBee tool. To ensure the annotation quality, three rounds of cross-checking and corrections were conducted. Specifically, the annotation content and criteria can be summarized as follows.

**Bounding Box Annotation of Hand Action:** We annotated the bounding boxes for both interaction regions and interaction objects. (1) *Interaction Regions (IR)*: We divided the hand's interaction region into three sub-interaction regions: left hand-object (L-O), right hand-object (R-O), and object-object (O-O) interaction regions, respectively. The L-O interaction region involves direct contact between the left hand and an object to perform an action (Figure 1 bottom left). Similarly, the R-O interaction region involves direct contact between the right hand and an object (Figure 1 bottom middle). The O-O interaction region indicates the contact between two objects (Figure 1 bottom right). (2) *Interaction Objects (IO)*: To better understand interaction actions, we also annotated the interactive object pair within each sub-interaction region using bounding boxes. For the L-O interaction region, we annotated left hand and left hand direct touching objects. Similarly, for the R-O interaction region, we annotated right hand and right hand direct touching objects. In the O-O interaction region, we annotated objects interact with each other in the context of a specific hand action (*i.e.*, *utility knife* and *carrot*). We also considered the *"active-passive"* relationship between objects, including the **active force provider** (*i.e.*, *utility knife*) and **passive force receiver** (*i.e.*, *carrot*), and annotate them in order in the triplet. However, in the annotation process, we may encounter overlapping bounding boxes, *i.e.*, the same interactive object will satisfy two annotation definitions, for example, the *utility knife* in Figure 1, which is both the object directly touched by the right hand in the R-O region and the active force provider in the O-O region. In this case, we annotate all the labels because the same object participates in different interaction actions in different interaction regions and has different roles (Annotation details can be seen in Appendix A.3.2). Finally, we annotated a total of 198,839 bounding boxes over 9 types, including 49,746 hand boxes, 66,402 interaction region boxes, and 82,691 interaction object boxes. Compared to existing datasets Damen et al. (2018), we added an average of 5 additional annotation types per frame.

**Hand Action Triplet Annotation:** We annotated fine-grained actions for each sub-interaction region. Unlike existing datasets, we represented each action in a triplet format: *<subject, action verb, object>*. The subject refers to the active force provider, the object refers to the passive force receiver, and the action verb describes the specific fine-grained hand action within the hand interaction region. (1) *Subject & Object*: In the L-O or R-O interaction regions, we labeled the subject as the corresponding hand and used fine-grained sub-categories for the interacting objects. To define the object noun, we referred to the EPIC-KITCHENS Damen et al. (2018) dataset. Furthermore, to enrich the description of each action, we included the contact areas of both the subject and object within the sub-interaction region. For example, in the L-O interaction region shown in Figure 1 bottom left, we labeled the subject as "*hand_left*" and the object as "*carrot_end*" based on their respective contact areas within the current interaction region. (2) *Action Verb*: We used fine-grained verbs in the annotated action triplets and constructed the verb vocabulary by sourcing from EPIC-KITCHENS Damen et al. (2018), AVA Gu et al. (2018), and Kinetics 700 Carreira et al. (2019).

**Object Segment Annotation:** To enrich our FHA-Kitchens, we utilized the state-of-the-art SAM model Kirillov et al. (2023) to annotate object masks in all video frames, which can be used for action segmentation relevant tasks.

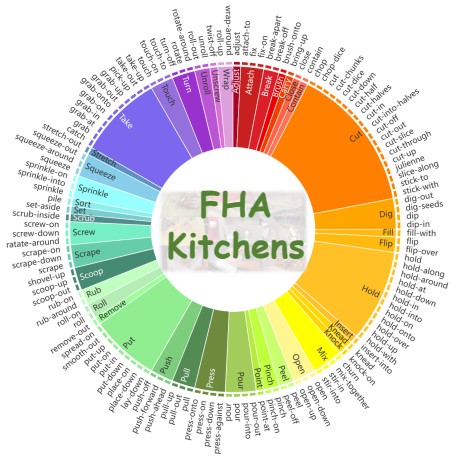

Figure 2: An overview of the action verbs and their parent action categories in FHA-Kitchens.

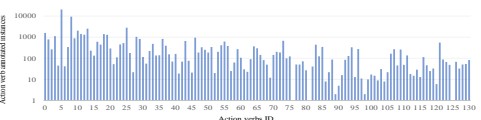

Figure 3: The distribution of instances per action verb category (the outer ring of the circle in Figure 2) in the FHA-Kitchens dataset.

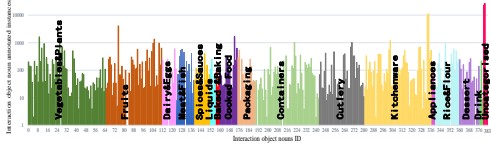

Figure 4: The distribution of instances per object noun category from 17 super-categories in the FHA-Kitchens dataset.

## 3.3 STATISTICS OF THE FHA-KITCHENS DATASET

**Overview of FHA-Kitchens.** As summarized in Table 1, we annotated action triplets for 30,047 frames from 2,377 clips, resulting in 878 action triplet categories, 131 action verbs and 384 interaction object nouns. We have taken steps to refine the dataset by focusing on hand actions and interaction regions, providing more fine-grained hand action categories and rich localization bounding boxes for the three sub-interaction regions (*i.e.*, L-O, R-O, and O-O). Compared to the original coarse-grained annotations in Kinetics 700_2020 Smaira et al. (2020), our dataset expanded the action labels by 7 dimensions, increased the number of action categories by 52 times, and introduced 123 new action verbs. Furthermore, we provide bounding boxes for hand action regions (*i.e.*, 66,402 interaction region boxes). This expansion significantly enhances the diversity of hand actions, provides valuable region-level contextual information for each action, and holds the potential to facilitate future research for a wider range of video understanding tasks. The FHA-Kitchens dataset was then randomly divided into disjoint train, validation, and test sets, with a video clip-based ratio of 7:1:2.

**Annotation Statistics.** Our annotation primarily focuses on hand interaction regions, interaction objects, and their corresponding interaction actions, resulting in a diverse array of verbs, nouns, and bounding boxes. Following the fine-grained annotation principles Damen et al. (2018), we ensured minimal semantic overlap among action verb-noun categories, rendering them suitable for multi-category action recognition and detection. **(1) Verbs:** The annotated dataset comprises 131 action verbs that have been grouped into 43 parent verb categories (Figure 2 and Figure 3). The three most prevalent parent verb categories, based on the count of sub-action verbs, are *Cut*, *Hold*, and *Take*, representing the most frequently occurring hand actions in human interactions. Figure 3 visually depicts the distribution of all verb categories within FHA-Kitchens, ensuring the presence of at least one instance for each verb category. **(2) Nouns:** In our annotation process, we identified a total of 384 interaction object noun categories that are associated with actions, categorized into 17 super-categories. Figure 4 shows the distribution of noun categories based on their affiliations with super-categories. Notably, the super-category "vegetables & plants" exhibits the highest number of sub-categories, followed by "kitchenware", which aligns with typical kitchen scenes. **(3) Bounding Boxes:** We performed a comprehensive statistical analysis on the bounding boxes of the three hand sub-interaction regions and the corresponding interaction objects. Specifically, we focused on two aspects: the box area and the aspect ratio. Detailed results can be found in Appendix A.2.

**Long-tail Property.** The distribution of instances per action triplet category in FHA-Kitchens, as depicted in Appendix A.2, depicts a long-tail property. This distribution reflects the frequency of hand interactions in real-world kitchen scenes, taking into account the varying commonness or rarity of specific hand actions. For instance, the action triplet "<hand_right, hold-in, utility-knife_handle>" consists of 9,887 instances, which is nine times more prevalent than the "<hand_left, hold-in, utility-knife_handle>" triplet. This long-tail characteristic of the distribution renders FHA-Kitchens a challenging benchmark for hand action recognition, making it suitable for investigating few-shot learning and out-of-distribution generalization in action recognition as well.

# 4 EXPERIMENT

## 4.1 IMPLEMENTATION DETAILS

We benchmark several representative action recognition methods Feichtenhofer et al. (2019); Liu et al. (2022); Wang et al. (2016; 2023); Ryali et al. (2023) and detection methods Ren et al. (2015); Ge et al. (2021); Zhu et al. (2020) with different backbone networks on the proposed FHA-Kitchens dataset based on the MMAction2 Contributors (2020) and MMDetection Chen et al. (2019) codebases. All models on the SL-D, SL-AR, and DG tracks are trained and tested using 4 NVIDIA GeForce RTX 3090 GPUs. For the SL-D and DG tracks, we employ the mean average precision (mAP) Lin et al. (2014) as the primary evaluation metric, while for the SL-AR track, Top-1 accuracy and Top-5 accuracy (%) are adopted.

## 4.2 SL-D TRACK: SUPERVISED LEARNING FOR HAND INTERACTION REGION AND OBJECT DETECTION

**Settings.** The SL-D track aims to evaluate the performance of different detection models on hand interaction regions and objects. We conducted experiments on the three representative methods, *i.e.*, Faster-RCNN Ren et al. (2015), YOLOX Ge et al. (2021), and Deformable DETR Zhu et al. (2020), with different backbones. Specifically, we pre-trained the model on the MS COCO Lin et al. (2014) object detection dataset and fine-tuned it on FHA-Kitchens for the tasks of hand interaction region detection and interaction object detection, respectively. For different models, we use the recommended optimization strategy (SGD or AdamW optimizer), initial learning rate, and batch size. The maximum training period is set to 100 epochs.

**Results on the SL-D Track.** The detection results are summarized in Table 2. As can be seen, detecting interaction objects is more difficult than detecting hand interaction regions, due to the fact that our interaction objects contain many small objects. This also validate the challenge posed by the diversity and fine-grained object categories in our dataset. Moreover, using a stronger backbone leads to slightly better detection results. It is noteworthy that, unlike existing action datasets, our FHA-Kitchens dataset provides abundant annotations of hand interaction regions, making it possible to investigate the model's ability to localize interaction regions and interpret hand actions in a more informative way, which is crucial for embodied intelligence research Li et al. (2023); Gupta et al. (2021). The visualization of the detection results can be found in the AppendixA.1.2.

Table 2: Detection results (mAP) of hand interaction regions and objects using different methods, *i.e.*, Faster-RCNN Ren et al. (2015), YOLOX Ge et al. (2021), and Deformable DETR Zhu et al. (2020), with different backbones on the validation set of the SL-D track. IR: Interaction Regions, IO: Interaction Objects.

|  | Backbone | IR | IO |
|---|---|---|---|
| *Two-stage methods* | | | |
| Faster-RCNN | ResNet50 | 65.20 | 40.80 |
| | ResNet101 | 66.10 | 41.90 |
| two-stage Deformable DETR | ResNet50 | 74.10 | 52.30 |
| *One-stage methods* | | | |
| YOLOX | YOLOX-s | 71.80 | 44.60 |
| | YOLOX-x | 75.60 | 49.00 |
| Deformable DETR Zhu et al. (2020) | ResNet50 | 73.00 | 53.00 |

Table 3: Classification results (Top-1 and Top-5 accuracy) of fine-grained hand actions using different features and the skeleton-based STGCN Yan et al. (2018) method (pre-trained on NTU60 Shahroudy et al. (2016)) on the validation set of the SL-AR.

| Feature | Top-1 | Top-5 |
|---|---|---|
| joint-2d | 22.78 | 47.68 |
| joint-3d | 22.36 | 52.32 |
| bone-2d | 22.36 | 49.79 |
| bone-3d | 24.05 | 52.32 |

## 4.3 SL-AR TRACK: SUPERVISED LEARNING FOR FINE-GRAINED HAND ACTION RECOGNITION

**Settings.** The SL-AR track primarily evaluates the performance of different action recognition models on fine-grained hand actions. We adopt the representative TSN Wang et al. (2016) and

Slowfast Feichtenhofer et al. (2019) with the ResNet50 and ResNet101 backbones, VideoSwin Liu et al. (2022) with the Swin-B backbone, VideoMAE V2 Wang et al. (2023) with the three different size backbones, and Hiera Ryali et al. (2023) with the Hiera-B backbone. We train these models on the FHA-Kitchens dataset using two settings: (1) **Pre-training on Kinetics 400 Carreira & Zisserman (2017) and hybrid dataset**, where we initialize the backbone with Kinetics 400 or Hybrid dataset pre-trained weights and fine-tune the entire model on the FHA-Kitchens training set; and (2) **Training from scratch on FHA-Kitchens**, where we randomly initialize the model weights and directly train them on FHA-Kitchens. For different models, we use the recommended optimization strategy and batch size and the maximum training period is set to 210 epochs.

**Results on the SL-AR Track.** The results in Table 4 show that the performance trends of all action recognition methods on FHA-Kitchens are similar to their performance on Kinetics 400. However, all the models achieve much worse accuracy on our FHA-Kitchens than the coarse-grained Kinetics 400, and unsatisfactory performance even using the large models. This is clear evidence that validates the challenging nature of the fine-grained hand action recognition on FHA-Kitchens. Besides, the utilization of pre-trained weights has proven to be beneficial, resulting in improved accuracy compared to training models from scratch. This finding suggests that despite the existence of a domain gap between coarse-grained and fine-grained actions, pre-training remains an effective strategy for addressing the challenges inherent in FHA-Kitchens, which has a larger number of action categories and relatively limited training data. In addition, we further supplemented the hand pose information and conducted experiments using the skeleton-based STGCN Yan et al. (2018) method. As shown in Table 3, 3D pose features outperform 2D pose features and bone features achieve better results than joint features (Please refer to the Appendix A.1.1 for more results analysis and analysis.)

Table 4: Classification results (Top-1 and Top-5 accuracy) of fine-grained hand actions using different methods on the validation set of the SL-AR track. w/ Pre-train: using pre-trained weights. w/o Pre-train: training from scratch.

| Method | Backbone | Pre-train Data | w/ Pre-train | | w/o Pre-train | |
|---|---|---|---|---|---|---|
| | | | Top-1 | Top-5 | Top-1 | Top-5 |
| TSN Wang et al. (2016) | ResNet50 | Kinetics 400 | 30.37 | 74.26 | 29.11 | 73.84 |
| | ResNet101 | Kinetics 400 | 30.80 | 73.42 | 30.38 | 74.26 |
| SlowFast Feichtenhofer et al. (2019) | ResNet50 | Kinetics 400 | 33.33 | 70.46 | 27.85 | 68.35 |
| | ResNet101 | Kinetics 400 | 36.71 | 67.93 | 31.22 | 69.62 |
| VideoSwin Liu et al. (2022) | Swin-B | Kinetics 400 | 37.13 | 70.89 | 34.18 | 66.67 |
| VideoMAE V2 Wang et al. (2023) | ViT-B | UnlabeledHybrid | 21.67 | 57.08 | - | - |
| | ViT-L | UnlabeledHybrid | 32.92 | 68.75 | - | - |
| | ViT-H | UnlabeledHybrid | 34.58 | 68.33 | - | - |
| Hiera Ryali et al. (2023) | Hiera-B | Kinetics 400 | 27.00 | 69.20 | - | - |

## 4.4 DG Track: Intra- and Inter-class Domain Generalization for Interaction Region Detection

### 4.4.1 Intra-class Domain Generalization

**Settings.** We conducted intra-class DG experiments using the three most prevalent parent action categories, *i.e.*, *Cut*, *Hold*, and *Take*. For each parent action category, we selected the most prevalent sub-categories and adopted the cross-validation protocol, *i.e.*, randomly choosing one sub-category as the test set while using all other sub-categories for training. Following the SL-D track, we selected the Faster RCNN Ren et al. (2015) model with the ResNet50 backbone as the default model, which is pre-trained on the MS COCO Lin et al. (2014) object detection dataset.

**Results on the Intra-class DG Track.** The results on *Cut* are summarized in Table 5, while the results on *Hold* and *Take* are shown in the Appendix A.1.1 due to the page limit. The performance of all four detection models remains stable for the sub-categories seen during training but deteriorates for unseen sub-categories, as evidenced by the diagonal scores, which exhibit a minimum drop of 15 mAP. This finding suggests that there is still potential for enhancing the models' generalization abilities, *e.g.*, by exploring the domain generalization or unsupervised domain adaptation techniques.

Table 5: Intra-class DG test results of Faster RCNN Ren et al. (2015) with the ResNet50 backbone on the "**Cut**" Setting. $\Delta_i = \frac{1}{3}\sum_{j,j\neq i} ji - ii, \Delta_i^* = \frac{1}{3}\sum_{j,j\neq i} ij - ii, i = 0,1,2,3.$

| Train | Test (mAP) | | | | $\Delta^*$ |
|---|---|---|---|---|---|
| | cut-slice | cut-off | cut-down | cut-dice | |
| w/o cut-slice | **33.30** | 65.00 | 56.00 | 60.90 | 27.33 |
| w/o cut-off | 57.10 | **48.00** | 54.80 | 62.80 | 10.23 |
| w/o cut-down | 57.30 | 64.40 | **41.30** | 63.50 | 20.43 |
| w/o cut-dice | 57.50 | 64.90 | 58.70 | **41.10** | 19.27 |
| $\Delta$ | 24.00 | 16.77 | 15.20 | 21.30 | |

Table 6: Inter-class DG test results. $\Delta_i = ii - \frac{1}{2}\sum_{j,j\neq i} ji, \Delta_i^* = ii - \frac{1}{2}\sum_{j,j\neq i} ij, i = 0,1,2.$

| Train | Test (mAP) | | | $\Delta^*$ |
|---|---|---|---|---|
| | Cut | Hold | Take | |
| Cut | **37.40** | 29.50 | 29.20 | 8.05 |
| Hold | 48.70 | **52.30** | 41.80 | 7.05 |
| Take | 14.00 | 13.20 | **41.20** | 27.60 |
| $\Delta$ | 6.05 | 30.95 | 5.70 | |

### 4.4.2 INTER-CLASS DOMAIN GENERALIZATION

**Settings.** We chose the three most prevalent parent action categories *Cut*, *Hold*, and *Take*, and adopted the cross-validation protocol, *i.e.*, randomly choosing one parent category for training and using the other parent categories for testing. Other settings follow those in the intra-class DG track.

**Results on the Inter-class DG Track.** The results are listed in Table 6. Similar to the results in the intra-class DG track, the detection models perform well on the seen categories while deteriorating on the unseen categories. Nevertheless, it is interesting to find that the performance gap ($\triangle_0 = 6.05$ and $\triangle_0^* = 8.05$) between *Cut* and others are smaller than those in the intra-class DG track, implying that there is likely a large intra-class variance, and the detection model is prone to overfitting the seen categories, particularly when the volume of training data is smaller (there are 7,463 training frames in *Hold* while only 1,680 in *Take*).

## 5 DISCUSSION

Through the SL-D, SL-AR, and DG tracks experiments, we have made a preliminary investigation of some unexplored research questions regarding fine-grained action detection and recognition. The obtained compelling empirical evidence not only highlights the inherent challenges associated with fine-grained hand action analysis but also reveals promising avenues for future research, demonstrating the value of the proposed FHA-Kitchens dataset. Our dataset may be slightly smaller in terms of the number of videos, but in terms of action granularity, we have expanded the action labels to 9 dimensions, resulting in 878 fine-grained hand action categories, which is 178 more than the number of categories in the large-scale dataset Kinetics700 Carreira et al. (2019). This provides a robust benchmark for fine-grained hand-action tasks. We will continue to increase the scale of our dataset. Our future research will also address the following aspects. Firstly, there is promising potential for further enhancing performance by delving deeper into the fine-grained categories, leveraging insightful ideas from the fields of fine-grained image analysis Wei et al. (2021) and action recognition Munro & Damen (2020). Secondly, considering the long-tail distribution of data in FHA-Kitchens, exploring strategies to balance the representation of both head and tail categories in the context of hand action detection and recognition warrants further investigation. Thirdly, while this paper focuses solely on closed-set detection and recognition tasks, exploring open-set settings holds both research and practical significance, which is supported by our FHA-Kitchens dataset. Lastly, the availability of mask annotations for all instances of hand interaction regions enables the study of action segmentation tasks, which can provide pixel-level interpretations of interaction objects.

## 6 CONCLUSION

In this paper, we build a novel dataset of fine-grained hand actions in kitchen scenes, i.e., FHA-Kitchens. The dataset offers a rich diversity in terms of viewpoints, occlusions, granularity, and action categories, providing new possibilities for fine-grained action research. We benchmark representative action recognition and detection methods on FHA-Kitchens and obtain compelling empirical evidence to understand the representation ability of different models, the impact of pre-training, the benefit of using diverse fine-grained hand actions for training, as well as intra- and inter-class domain generalization. We anticipate that FHA-Kitchens would pave the way for future research in this field.

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

# A APPENDIX

## A.1 MORE QUANTITATIVE AND QUALITATIVE RESULTS

### A.1.1 QUANTITATIVE RESULTS

**SL-AR Track results**. Table 7 presents the performance of different action recognition methods on the Kinetics 400 Carreira & Zisserman (2017) dataset and the proposed FHA-Kitchens dataset, with and without pre-trained models. From the experimental results, it can be observed that the performance trends of all action recognition methods on FHA-Kitchens are similar to their performance on Kinetics 400 Carreira & Zisserman (2017), while the models perform much better on the coarse-grained actions of Kinetics 400. For the best-performing VideoSwin Liu et al. (2022) model, the top-1 accuracy on Kinetics 400 surpasses the top-1 accuracy on FHA-Kitchens by 43.44%. And those methods with even large models cannot achieve satisfactory performance. This is clear evidence that validates the challenging nature of the fine-grained hand action recognition on FHA-Kitchens. Besides, the utilization of pre-trained weights has proven to be beneficial, resulting in improved accuracy compared to training models from scratch. This finding suggests that despite the existence of a domain gap between coarse-grained and fine-grained actions, pre-training remains an effective strategy for addressing the challenges inherent in FHA-Kitchens, which have a larger number of action categories and relatively limited training data.

Table 7: Classification results (Top-1 and Top-5 accuracy) of fine-grained hand actions using different methods on the validation set of the SL-AR track. w/ Pre-train : using pre-trained weights. w/o Pre-train: Training from scratch (the Kinetics 400 Carreira & Zisserman (2017) dataset results from mmaction2 Contributors (2020), VideoMAE V2 Wang et al. (2023), and Hiera Ryali et al. (2023)).

| Dataset | Method | Backbone | Pre-train Data | w/ Pre-train | | w/o Pre-train | |
|---|---|---|---|---|---|---|---|
| | | | | Top-1 | Top-5 | Top-1 | Top-5 |
| Kinetics 400 | TSN Wang et al. (2016) | ResNet50 | ImageNet | 72.83 | 90.65 | - | - |
| | | ResNet101 | ImageNet | 75.89 | 92.07 | - | - |
| | SlowFast Feichtenhofer et al. (2019) | ResNet50 | - | - | - | 76.65 | 92.86 |
| | | ResNet101 | - | - | - | 78.65 | 93.88 |
| | VideoSwin Liu et al. (2022) | Swin-B | ImageNet | 80.57 | 94.49 | - | - |
| | VideoMAE V2 Wang et al. (2023) | ViT-B | UnlabeledHybrid | 81.50 | - | - | - |
| | | ViT-L | UnlabeledHybrid | 85.40 | - | - | - |
| | | ViT-H | UnlabeledHybrid | 86.90 | - | - | - |
| | Hiera Ryali et al. (2023) | Hiera-B | Kinetics 400 | 84.00 | - | - | - |

| Dataset | Method | Backbone | Pre-train Data | w/ Pre-train | | w/o Pre-train | |
|---|---|---|---|---|---|---|---|
| | | | | Top-1 | Top-5 | Top-1 | Top-5 |
| FHA-Kitchens | TSN Wang et al. (2016) | ResNet50 | Kinetics 400 | 30.37 | 74.26 | 29.11 | 73.84 |
| | | ResNet101 | Kinetics 400 | 30.80 | 73.42 | 30.38 | 74.26 |
| | SlowFast Feichtenhofer et al. (2019) | ResNet50 | Kinetics 400 | 33.33 | 70.46 | 27.85 | 68.35 |
| | | ResNet101 | Kinetics 400 | 36.71 | 67.93 | 31.22 | 69.62 |
| | VideoSwin Liu et al. (2022) | Swin-B | Kinetics 400 | 37.13 | 70.89 | 34.18 | 66.67 |
| | VideoMAE V2 Wang et al. (2023) | ViT-B | UnlabeledHybrid | 21.67 | 57.08 | - | - |
| | | ViT-L | UnlabeledHybrid | 32.92 | 68.75 | - | - |
| | | ViT-H | UnlabeledHybrid | 34.58 | 68.33 | - | - |
| | Hiera Ryali et al. (2023) | Hiera-B | Kinetics 400 | 27.00 | 69.20 | - | - |

In addition, we further supplemented the hand pose information and conducted experiments using the skeleton-based STGCN Yan et al. (2018) method. We used STGCN pre-trained on NTU60 Shahroudy et al. (2016) and NTU120 Liu et al. (2019) and fine-tuned the models on the SL-AR track using different features for fine-grained hand actions, the results (Top-1 and Top-5 accuracy) can be seen in Table 8.

Table 8: Classification results (Top-1 and Top-5 accuracy) of fine-grained hand actions using different features and the skeleton-based STGCN Yan et al. (2018) method (pre-trained on NTU60 Shahroudy et al. (2016) and NTU120 Liu et al. (2019)) on the validation set of the SL-AR track.

| Pre-train Data | Feature | Top-1 | Top-5 | Pre-train Data | Feature | Top-1 | Top-5 |
|---|---|---|---|---|---|---|---|
| NTU60 | joint-2d | 22.78 | 47.68 | NTU120 | joint-2d | 20.68 | 48.10 |
| | joint-3d | 22.36 | 52.32 | | joint-3d | 21.10 | 47.68 |
| | joint-motion-2d | 8.02 | 19.83 | | joint-motion-2d | 9.28 | 20.25 |
| | joint-motion-3d | 10.97 | 23.63 | | joint-motion-3d | 11.81 | 26.16 |
| | bone-2d | 22.36 | 49.79 | | bone-2d | 24.05 | 57.81 |
| | bone-3d | 24.05 | 52.32 | | bone-3d | 24.05 | 51.05 |
| | bone-motion-2d | 10.55 | 23.21 | | bone-motion-2d | 9.28 | 23.21 |
| | bone-motion-3d | 13.50 | 26.16 | | bone-motion-3d | 12.24 | 27.00 |

According to the experimental results, it can be observed that 3D pose features outperform 2D pose features and bone features achieve better results than joint features. Nevertheless, the overall results did not surpass the efficacy of hand-object interaction-based approaches, highlighting that relying only on hand pose information is insufficient for accomplishing fine-grained action recognition tasks. Because the generation of hand actions involves interacting objects, achieving a fine-grained hand action recognition task is required to consider the information of the objects interacting with the hand, which is different from a whole-body action recognition task (e.g., AVA, FineGym dataset).

Note that the main difficulty of FHA-Kitchens lies in fine-grained action recognition rather than localization. Figure 5 shows some examples of high-scoring localization but false recognition instances, demonstrating that the difficulty of recognition lies in fine-grained details.

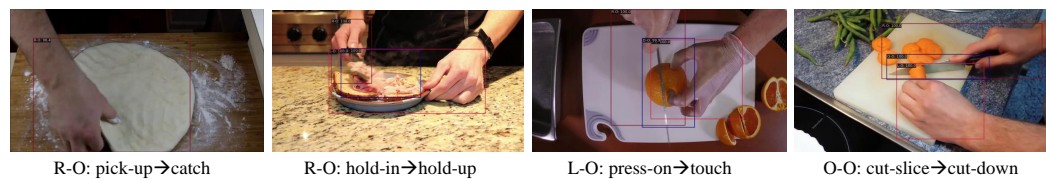

R-O: pick-up→catch    R-O: hold-in→hold-up    L-O: press-on→touch    O-O: cut-slice→cut-down

Figure 5: Some examples of high-scoring localization but false recognition instances. The recognition model often struggles to discriminate fine-grained details.

**DG Track results**. Here, we present the test results of two additional parent category actions, *i.e.*, *Hold* and *Take*, within the Intra-class DG track (Section 4.4.1 of the paper). It is evident from the results that the performance on sub-categories seen during training surpasses that of unseen sub-categories. This discrepancy is supported by the diagonal scores, which reveal a significant decline of up to 46 mAP. These findings align with the observations in Section 4.4.1 of the paper, emphasizing the existence of large intra-class variance and underscoring the need for more efforts to improve the generalization abilities of the models.

Table 9: Intra-class DG test results of Faster RCNN Ren et al. (2015) with the ResNet50 backbone on the "**Hold**" Setting. $\Delta_i = \frac{1}{2} \sum_{j,j \neq i} ji - ii, \Delta_i^* = \frac{1}{2} \sum_{j,j \neq i} ij - ii, i = 0, 1, 2.$

| Train | Test (mAP) | | | $\Delta^*$ |
|---|---|---|---|---|
| | hold-up | hold-in | hold-around | |
| w/o hold-up | **44.00** | 53.50 | 71.70 | 18.60 |
| w/o hold-in | 44.30 | **7.30** | 69.10 | 49.40 |
| w/o hold-around | 52.30 | 52.80 | **47.80** | 4.75 |
| $\Delta$ | 4.30 | 45.85 | 22.60 | |

Table 10: Intra-class DG test results of Faster RCNN Ren et al. (2015) with the ResNet50 backbone on the "**Take**" Setting. $\Delta_i = \frac{1}{2}\sum_{j,j\neq i} ji - ii, \Delta_i^* = \frac{1}{2}\sum_{j,j\neq i} ij - ii, i = 0, 1, 2.$

|  | Test (mAP) | | | |
| --- | --- | --- | --- | --- |
| Train | pick-up | grab | catch | $\Delta^*$ |
| w/o pick-up | **0.40** | 47.10 | 46.60 | 46.45 |
| w/o grab | 19.00 | **4.50** | 39.00 | 24.50 |
| w/o catch | 19.10 | 46.00 | **15.60** | 16.95 |
| $\Delta$ | 18.65 | 42.05 | 27.20 | |

### A.1.2 QUALITATIVE RESULTS

The visual results of the SL-D and SL-AR track experiments are presented in Figure 13, Figure 14, and Figure 15. We showcase the visualizations for interaction region detection, interaction object detection, and action recognition, focusing on various interaction scenarios that vary in the complexity of hand interaction regions. In the recognition results, we provide fine-grained action verbs corresponding to the three hand interaction regions, denoted as *<L-O action verb, R-O action verb, O-O action verb>*. Figure 13 shows some challenging cases of hand interactions, providing compelling evidence of the good prediction performance of detection and recognition models, *i.e.*, the Faster-RCNN Ren et al. (2015) with a ResNet50 backbone for detection and a pre-trained TSN Wang et al. (2016) model with a ResNet50 backbone for action recognition. Moreover, Figure 14 and Figure 15 also demonstrate accurate detection and recognition results for some common interaction cases.

### A.2 MORE STATISTICS OF THE FHA-KITCHENS DATASET

In this part, we re-arrange some figures in the paper to make them more readable and provide more statistics of the FHA-Kitchens dataset. Our annotation primarily focuses on hand interaction regions, interaction objects, and their corresponding interaction actions, resulting in a diverse array of verbs, nouns, and bounding boxes.

- **Verbs:** The annotated dataset comprises 131 action verbs that have been grouped into 43 parent verb categories (Figure 6 and Figure 7). The three most prevalent parent verb categories, based on the count of sub-action verbs, are *Cut*, *Hold*, and *Take*, representing the most frequently occurring hand actions in human interactions. Figure 7 visually depicts the distribution of all verb categories within FHA-Kitchens, ensuring the presence of at least one instance for each verb category. Specifically, the mapping between action verb IDs and their corresponding category names can be seen in Table 12.

- **Nouns:** In our annotation process, we identified a total of 384 interaction object noun categories that are associated with actions, categorized into 17 super-categories. Figure 16 shows the distribution of noun categories based on their affiliations with super-categories. Notably, the super-category "vegetables & plants" exhibits the highest number of sub-categories, followed by "kitchenware", which aligns with typical kitchen scenes. Specifically, the mapping between interaction object noun IDs and their corresponding category names can be seen in Table 13, Table 14, and Table 15.

- **Bounding Boxes:** We performed a comprehensive statistical analysis on the bounding boxes of the three hand interaction regions and the corresponding interaction objects. Specifically, we focused on two aspects: the box area and the aspect ratio. Detailed results can be found in Figure 8 and Figure 9. Figure 8 shows the considerable range of sizes covered by our bounding boxes, with many interaction objects exhibiting small and challenging sizes for accurate detection. Moreover, in Figure 9, the aspect ratios of the bounding boxes exhibit notable variation. The aspect ratios of the three regions tend to concentrate within the range of [0.5,2], which can be attributed to the typical composition of interaction regions involving two interacting objects. Consequently, the bounding box encompasses the combined region of both objects. For instance, the R-O interaction region frequently involves the interaction between the "right hand" and "utility knife". In such cases, the aspect ratio of the bounding

box is observed to be 2:1, as depicted in Figure 11. These findings highlight the significant challenges of the detection task in our dataset.

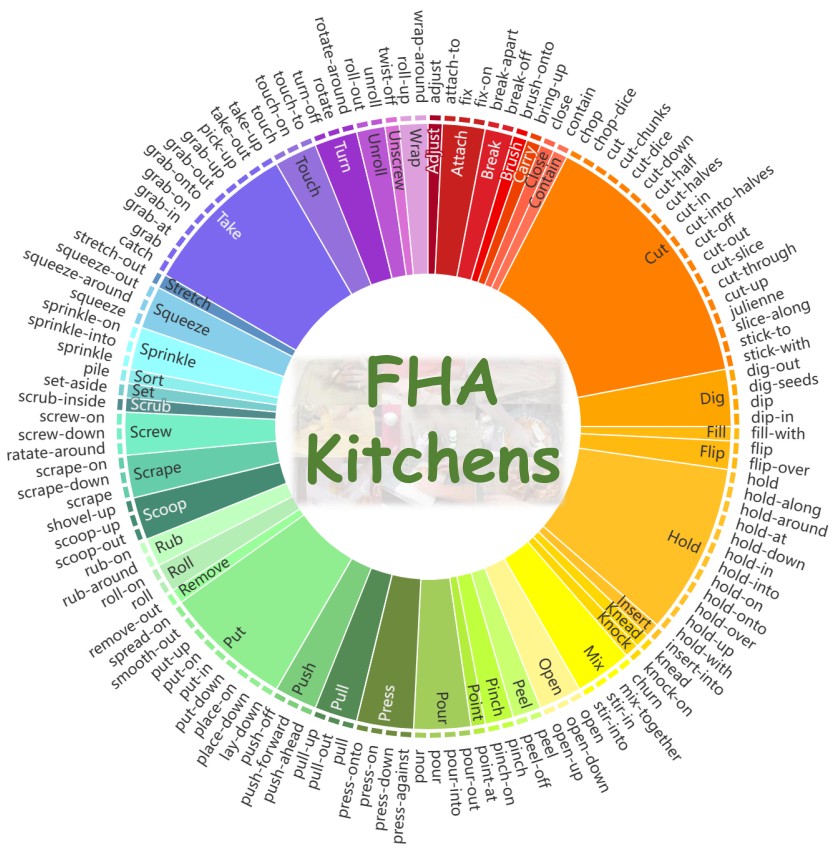

Figure 6: An overview of the action verbs and their parent action categories in FHA-Kitchens.

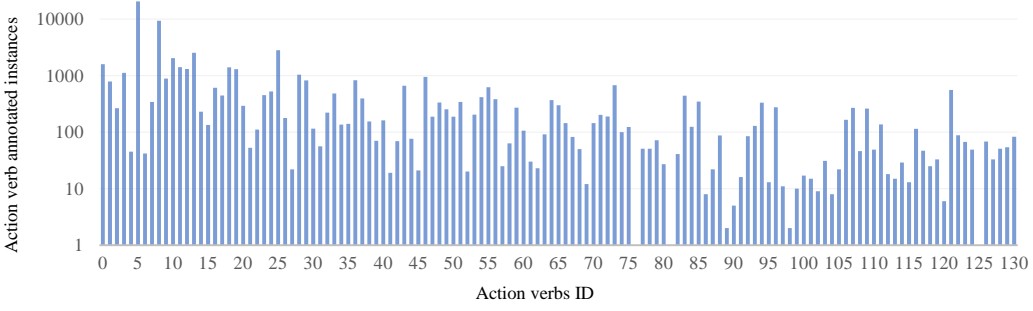

Figure 7: The distribution of instances per action verb category (the outer ring of the circle in Figure 6) in the FHA-Kitchens dataset.

**Long-tail Property.** The distribution of instances per action triplet category in FHA-Kitchens, as depicted in Figure 10, depicts a long-tail property. This distribution reflects the frequency of hand interactions in real-world kitchen scenes, taking into account the varying commonness or rarity of specific hand actions. For instance, the action triplet "<hand_right, hold-in, utility-knife_handle>" consists of 9,887 instances, which is nine times more prevalent than the "<hand_left, hold-in, utility-knife_handle>" triplet. This long-tail characteristic of the distribution renders FHA-Kitchens a challenging benchmark for hand action recognition, making it suitable for investigating few-shot learning and out-of-distribution generalization in action recognition as well.

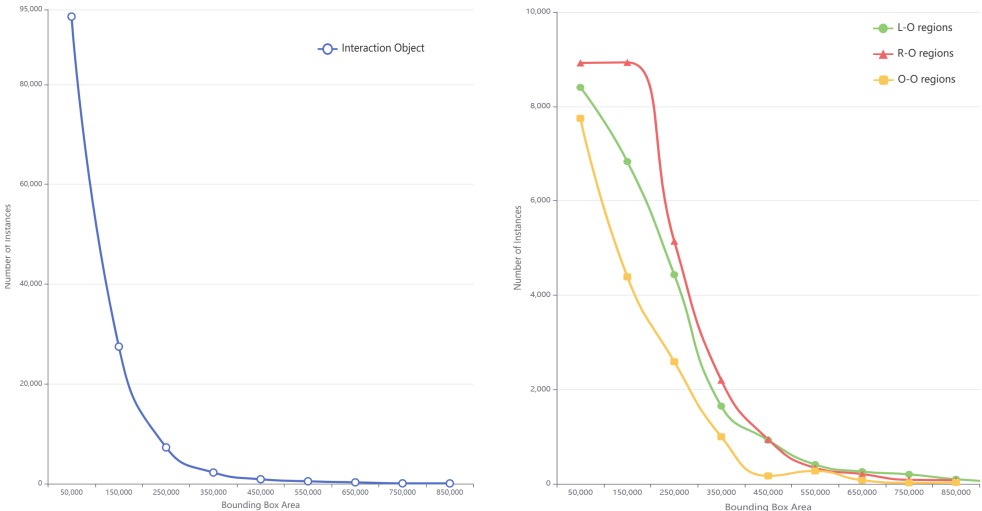

Figure 8: The distributions of bounding box areas of interaction objects (left) and interaction regions (right) in the FHA-Kitchens dataset.

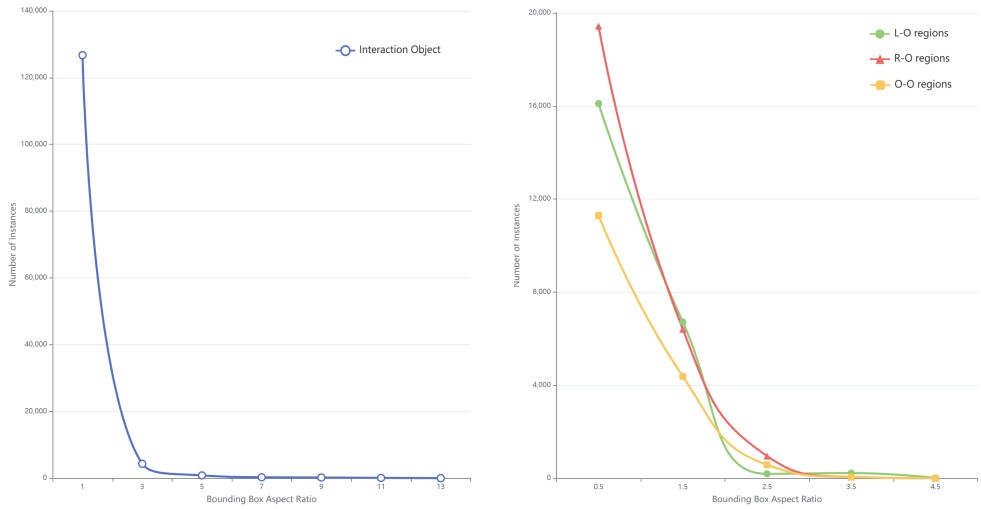

Figure 9: The distributions of bounding box aspect ratios of interaction objects (left) and interaction regions (right) in the FHA-Kitchens dataset.

### A.3 DATASHEETS FOR DATASETS

#### A.3.1 MOTIVATION

**1. For what purpose was the dataset created? Was there a specific task in mind? Was there a specific gap that needed to be filled? Please provide a description.**

**A1:** FHA-Kitchens is created to facilitate research in the field of fine-grained hand action recognition. It is important to study several challenging questions in the context of more training data from diverse fine-grained hand actions, such as: 1) How do different representative action recognition models perform on fine-grained hand action tasks? 2) How do state-of-the-art detection models perform on the refined hand interaction regions and interaction objects? 3) How about the impact of pre-training, e.g., on the full-body actions dataset Carreira & Zisserman (2017), in the context of the large-scale dataset with diverse fine-grained hand actions? and 4) How do the intra-class and

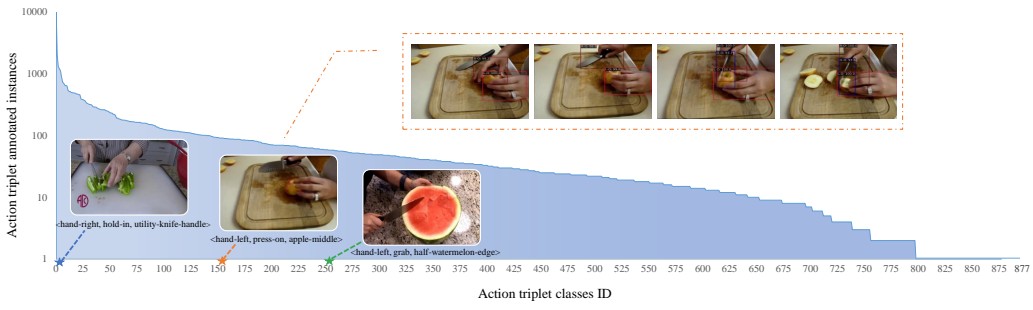

Figure 10: The distribution of instances per action triplet category in the FHA-Kitchens dataset.

inter-class generalization capabilities of models trained with specific fine-grained hand actions or parent hand actions perform? However, existing action datasets primarily focus on full-body human actions and lack emphasis on hand actions, with limited granularity in their treatment. Therefore, it is impossible to study these questions using existing datasets. In contrast, FHA-Kitchens primarily focuses on hand actions and refines hand interaction regions into three sub-interaction regions. Each sub-interaction region action was annotated using a triplet format, denoted as <subject, action verb, object>. Overall, the action information is expanded to nine dimensions, significantly enhancing the granularity of actions and providing valuable resources for researchers to study these questions effectively.

FHA-Kitchens aims to provide a better, more comprehensive, and finer-grained benchmark for hand action recognition. While there are existing datasets available for hand action recognition, they only cover limited hand action information and lack an in-depth understanding of hand actions. With its diverse and fine-grained hand action information, the FHA-Kitchens dataset enables a better evaluation of performance for fine-grained hand action recognition methods.

**2. Who created this dataset (e.g., which team, research group) and on behalf of which entity (e.g., company, institution, organization)?**

**A2:** Our dataset is created by the authors as well as some volunteer undergraduate students.

**3. Who funded the creation of the dataset? If there is an associated grant, please provide the name of the grantor and the grant name and number.**

**A3:** This information will be made public once the paper is accepted after peer review.

### A.3.2 COMPOSITION

**1. What do the instances that comprise the dataset represent (e.g., documents, photos, people, countries)? Are there multiple types of instances(e.g., movies, users, and ratings; people and interactions between them; nodes and edges)? Please provide a description.**

**A1:** FHA-Kitchens consists of video frames, including 878 fine-grained hand interaction action triplets. It primarily focuses on fine-grained actions generated within hand interaction regions, such as *cut-slice* and *hold-in*. For each frame, we provide bounding boxes for three hand sub-interaction regions (*i.e.*, left hand-object (L-O), right hand-object (R-O), and object-object (O-O) interaction regions) and the interaction objects. Each sub-interaction region action was annotated using a triplet format, denoted as *<subject, action verb, object>*. Additionally, we provide segmentation masks related to hands and interaction objects.

**2. How many instances are there in total (of each type, if appropriate)?**

**A2:** The FHA-Kitchens contains 30,047 frames from 2,377 video clips, with each frame annotated for three fine-grained hand interaction regions, resulting in a total of 878 fine-grained action triplets. Among them, there are 597 frames where no hand interaction action occurs, represented as L-O_triplet:<none>, R-O_triplet:<none>, O-O_triplet: <none>.

**3. Does the dataset contain all possible instances or is it a sample (not necessarily random) of instances from a larger set? If the dataset is a sample, then what is the larger set? Is the sample representative of the larger set (e.g., geographic coverage)? If so, please describe how this representativeness was validated/verified. If it is not representative of the larger set, please describe why not (e.g., to cover a more diverse range of instances, because instances were withheld or unavailable).**

**A3:** FHA-Kitchens is a real-world sample of human hands part in the kitchen scenes, including information about their actions. The data is sourced from an existing large-scale full-body action dataset Smaira et al. (2020), from which we selected videos featuring hand interaction actions. We extracted a total of 30 videos, amounting to 84.22 minutes of footage, encompassing 8 distinct types of dishes. Due to the diversity of real-world human hand actions, it's impossible to cover all types of actions. The FHA-Kitchens dataset focuses primarily on fine-grained tasks related to hand actions. To address the granularity issue, we improved the action information in the existing dataset. Compared to the data's original annotations in Kinetics-700_2020 Smaira et al. (2020), our dataset expanded the action labels by 7 dimensions, increased the number of action categories by 52 times, and introduced 123 new action verbs. We provide a finer-grained set of hand action instances than ever before, facilitating further research in fine-grained hand action recognition.

**4. What data does each instance consist of? "Raw" data (e.g., unprocessed text or images) or features? In either case, please provide a description.**

**A4:** Each instance consists of at most 9 kinds of bonding boxes (*i.e.*, three hand sub-interaction regions and interaction objects within interaction region) and sub-interaction region corresponding triplet descriptions (i.e., <*subject, action verb, object*>). Additionally, we took into account the "active-passive" relationships between object pairs and the specific contact areas involved in the interaction actions. Consequently, our annotation process encompassed a total of nine dimensions, resulting in 878 annotated action triplets for hand actions. The annotation details are listed in Figure12, and the corresponding visualizations are shown in Figure11.

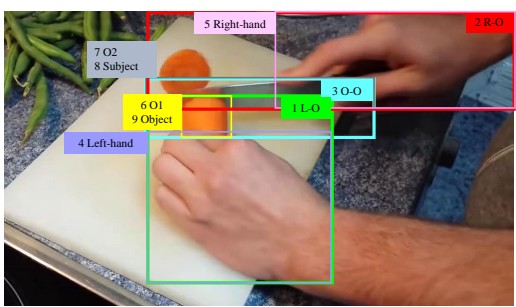

| | Bounding Box Annotation | | Action Triplets Annotation |
|---|---|---|---|
| *id* | *definition* | *b-box label* | *action triplet label* |
| 1 | left hand-object interaction region | L-O | (hand_left, press-on, carrot_end) |
| 2 | right hand-object interaction region | R-O | (hand_right, hold-in, utility-knife_handle) |
| 3 | object-object interaction region | O-O | (utility-knife_body, cut-slice, carrot_head) |
| 4 | left hand | Left-hand | - |
| 5 | right hand | Right-hand | - |
| 6 | object touched by left hand in L-O | O1 | - |
| 7 | object touched by right hand in R-O | O2 | - |
| 8 | active force provider in O-O | Subject | - |
| 9 | passive force receiver in O-O | Object | - |

Figure 11: Visualization of bounding box annotations for the example of "*fry vegetables*".

Figure 12: Descriptive list of action triplets and bounding box annotations.

**5. Is there a label or target associated with each instance? If so, please provide a description.**

**A5:** Yes. Due to our parallel annotation process, we generated annotation files in different styles. However, we consolidated all the bounding box and triplet annotation information into a single CSV file. In the merged CSV file, each instance is annotated with labels following the style of the Kinetics Carreira & Zisserman (2017); Carreira et al. (2018; 2019); Smaira et al. (2020) and AVA Gu et al. (2018) datasets, which include video_name, video_id, clip_id, clip_name, frame_name, timestamp, L-O_triplet, L-O_action_verb_id, L-O_action_verb_class, L-O_action_bbox, left_hand_bbox, O1_class, O1_bbox , R-O_triplet, R-O_action_verb_id, R-O_action_verb_class, R-O_action_bbox, right_hand_bbox, O2_class, O2_bbox, O-O_triplet, O-O_action_verb_id, O-O_action_verb_class, O-O_action_bbox, subject_class, subject_bbox, object_class, object_bbox, action_verb_triplet, action_verb_triplet_id.

**6. Is any information missing from individual instances? If so, please provide a description, explaining why this information is missing (e.g., because it was unavailable). This does not include intentionally removed information, but might include, e.g., redacted text.**

**A6:** Yes. Some instances may not have all nine types of bonding boxes and their corresponding action triplets and segmentation annotation because of severe occlusion, truncation, blur, or small scale. We just annotated "None" in our annotation file to represent this situation.

**7. Are relationships between individual instances made explicit (e.g., users' movie ratings, social network links)? If so, please describe how these relationships are made explicit.**

**A7:** Yes. We provide different styles of annotations files, in COCO-style, the annotations are connected by image id and category id, you can easily access them by COCO APIs. In CSV-style, one line represents the annotations of one frame and can be processed by the pandas library easily.

**8. Are there recommended data splits (e.g., training, development/validation, testing)? If so, please provide a description of these splits, explaining the rationale behind them.**

**A8:** Yes. We randomly split the dataset into the disjoint train, validation, and test sets following the ratio of 7:1:2.

**9. Are there any errors, sources of noise, or redundancies in the dataset? If so, please provide a description.**

**A9:** Although we conducted three rounds of cross-checking and corrections, there may still be some errors in the annotations, *e.g.*, inappropriate bounding box annotations or small drifts of the bounding box locations, incorrectly written verbs or nouns, insufficient granularity in verb or noun descriptions, inappropriate formatting of triplets, etc. However, we have made every effort to minimize such occurrences.

To analyze the quality of annotations, we randomly selected 500 frames and conducted manual evaluations for correctness. The results are reported in Table 11. These error rates are comparable to recently published datasets Damen et al. (2018).

Table 11: Error rate in FHA-Kitchens. I-O: Interaction Objects, I-R: Interaction Regions.

|  | Frames | I-O Boxes | I-R Boxes | Verb | Noun |
|---|---|---|---|---|---|
| **Total Number** | 500 | 3,006 | 1,503 | 1,503 | 2,006 |
| **Error Rate (%)** | - | 4.9 | 2.5 | 2.2 | 5.3 |

**10. Is the dataset self-contained, or does it link to or otherwise rely on external resources (e.g., websites, tweets, other datasets)? If it links to or relies on external resources, a) are there guarantees that they will exist, and remain constant, over time; b) are there official archival versions of the complete dataset (i.e., including the external resources as they existed at the time the dataset was created); c) are there any restrictions (e.g., licenses, fees) associated with any of the external resources that might apply to a future user? Please provide descriptions of all external resources and any restrictions associated with them, as well as links or other access points, as appropriate.**

**A10:** Our dataset was derived from a large-scale publicly available dataset, namely Kinetics-700_2020 Smaira et al. (2020), which is publicly available for download from their website. The Kinetics dataset follows the Creative Commons Attribution 4.0 International License. We would like to express our gratitude to the authors for their significant contributions to the research community.

**11. Does the dataset contain data that might be considered confidential (e.g., data that is protected by legal privilege or by doctorpatient confidentiality, data that includes the content of individuals non-public communications)? If so, please provide a description.**

**A11:** No.

**12. Does the dataset contain data that, if viewed directly, might be offensive, insulting, threatening, or might otherwise cause anxiety? If so, please describe why.**

**A12:** No.

### A.3.3    COLLECTION PROCESS

**1. How was the data associated with each instance acquired? Was the data directly observable (e.g., raw text, movie ratings), reported by subjects (e.g., survey responses), or indirectly inferred/derived from other data (e.g., part-of-speech tags, model-based guesses for age or language)? If data was reported by subjects or indirectly inferred/derived from other data, was the data validated/verified? If so, please describe how.**

**A1:** The FHA-Kitchens dataset follows the annotation styles of Kinetics-700_2020 Smaira et al. (2020) and AVA Gu et al. (2018) datasets, integrating all the annotation information (bounding box and triplet) into a single CSV file. Each row in the file represents the annotations for a single frame, and these annotations are visually accessible within the labels.

**2. What mechanisms or procedures were used to collect the data (e.g., hardware apparatus or sensor, manual human curation, software program, software API)? How were these mechanisms or procedures validated?**

**A2:** The data in FHA-Kitchens come from dataset publicly available datasets described above, which can be directly downloaded from their websites.

**3. If the dataset is a sample from a larger set, what was the sampling strategy (e.g., deterministic, probabilistic with specific sampling probabilities)?**

**A3:** Currently, we focus exclusively on hand interaction actions in kitchen scenes, thus primarily extracting data that includes hand interaction actions in kitchen scenes.

**4. Who was involved in the data collection process (e.g., students, crowdworkers, contractors) and how were they compensated (e.g., how much were crowdworkers paid)?**

**A4:** The first two authors collected this dataset. The annotation compensation is based on the prevailing market rates.

**5. Over what timeframe was the data collected? Does this timeframe match the creation timeframe of the data associated with the instances (e.g., recent crawl of old news articles)? If not, please describe the timeframe in which the data associated with the instances was created.**

**A5**: It took about 1 week to collect the data and about 6 weeks to complete organization and annotation, as each participant labeled the bonding boxes and action triplets about four hours per workday. And the segmentation masks are generated by the Segment-Anything Model Kirillov et al. (2023) guided by the bonding boxes, and corrected by human annotators for about one week.

### A.3.4    PREPROCESSING/CLEANING/LABELING

**1. Was any preprocessing/cleaning/labeling of the data done (e.g., discretization or bucketing, tokenization, part-of-speech tagging, SIFT feature extraction, removal of instances, processing of missing values)? If so, please provide a description. If not, you may skip the remainder of the questions in this section.**

**A1:** Yes. Since we focus on hand actions, we performed filtering and processing operations on the original videos, including the following three steps. (1) First, we observed that kitchen scenes often featured hand actions, with video content prominently showcasing human hand parts. Therefore, we sought out and extracted relevant videos that were set against a kitchen backdrop. (2)Then, to ensure the quality of the dataset, we selectively chose videos with higher resolutions. Specifically, 87% of the videos were recorded at $1,280 \times 720$ resolution, while another 13% had a shorter side of 480. Additionally, 67% of the videos were captured at 30 frames per second (fps), and another 33% were recorded at 24~25 fps. (3) Subsequently, we imposed a duration constraint on the videos, ranging from 30 seconds to 5 minutes, to exclude excessively long-duration videos. This constraint aimed to maintain a balanced distribution within the sample space. Finally, we collected a total of 30 videos, amounting to 84.22 minutes of footage, encompassing 8 distinct types of dishes.

The collected video data was reorganized and cleaned to align with our annotation criteria. First, we split the collected video data into individual frames, as our annotated units are frames. Subsequently, we conducted further cleaning of the frames by excluding those that did not depict hands or exhibited meaningless hand actions. This cleaning process took into consideration factors such as occlusion,

frame quality (*i.e.*, without significant blur, subtitles, and logos), meaningful hand actions, and frame continuity. As a result, we obtained a total of 30,047 high-quality candidate video frames containing diverse hand actions for our FHA-Kitchens dataset. Compared to the initial collection, 113,436 frames were discarded during the cleaning process.

We recruited 10 voluntary annotators, whose responsibility was to annotate fine-grained action triplet classes and bounding boxes for each hand interaction region. In order to enhance annotation efficiency, we implemented a parallel annotation pipeline. The annotation of action triplets was carried out on the Amazon Mechanical Turk platform, while the bounding box annotation was facilitated using the LabelBee tool. To ensure the annotation quality, three rounds of cross-checking and corrections were conducted.

**2. Was the "raw" data saved in addition to the preprocessed/cleaned/labeled data (e.g., to support unanticipated future uses)? If so, please provide a link or other access point to the "raw" data.**

**A2:** No.

**3. Is the software used to preprocess/clean/label the instances available? If so, please provide a link or other access point.**

**A3:** The annotation of action triplets was carried out on the Amazon Mechanical Turk platform, while the bounding box annotation was facilitated using the LabelBee tool.

### A.3.5 USES

**1. Has the dataset been used for any tasks already? If so, please provide a description.**

**A1:** No.

**2. Is there a repository that links to any or all papers or systems that use the dataset? If so, please provide a link or other access point.**

**A2:** N/A.

**3. What (other) tasks could the dataset be used for?**

**A3:** FHA-Kitchens can be used for the research of fine-grained hand action recognition and hand interaction region and object detection. Besides, it can also be used for specific machine learning topics such as domain generalization and action segmentation. Please see the Discussion part of the paper.

**4. Is there anything about the composition of the dataset or the way it was collected and preprocessed/cleaned/labeled that might impact future uses? For example, is there anything that a future user might need to know to avoid uses that could result in unfair treatment of individuals or groups (e.g., stereotyping, quality of service issues) or other undesirable harms (e.g., financial harms, legal risks) If so, please provide a description. Is there anything a future user could do to mitigate these undesirable harms?**

**A4:** No.

**5. Are there tasks for which the dataset should not be used? If so, please provide a description.**

**A5:** No.

### A.3.6 DISTRIBUTION

**1. Will the dataset be distributed to third parties outside of the entity (e.g., company, institution, organization) on behalf of which the dataset was created? If so, please provide a description.**

**A1:** Yes. The dataset will be made publicly available to the research community.

**2. How will the dataset will be distributed (e.g., tarball on website, API, GitHub)? Does the dataset have a digital object identifier (DOI)?**

**A2:** It will be publicly available on the dataset project website at GitHub.

**3. When will the dataset be distributed?**

**A3:** The dataset will be distributed once the paper is accepted after peer review.

**4. Will the dataset be distributed under a copyright or other intellectual property (IP) license, and/or under applicable terms of use (ToU)? If so, please describe this license and/or ToU, and provide a link or other access point to, or otherwise reproduce, any relevant licensing terms or ToU, as well as any fees associated with these restrictions.**

**A4:** It will be distributed under the MIT license.

**5. Have any third parties imposed IP-based or other restrictions on the data associated with the instances? If so, please describe these restrictions, and provide a link or other access point to, or otherwise reproduce, any relevant licensing terms, as well as any fees associated with these restrictions.**

**A5:** No.

**6. Do any export controls or other regulatory restrictions apply to the dataset or to individual instances? If so, please describe these restrictions, and provide a link or other access point to, or otherwise reproduce, any supporting documentation.**

**A6:** No.

### A.3.7 MAINTENANCE

**1. Who will be supporting/hosting/maintaining the dataset?**

**A1:** The authors.

**2. How can the owner/curator/manager of the dataset be contacted (e.g., email address)?**

**A2:** They can be contacted via email available on the our dataset project website.

**3. Is there an erratum? If so, please provide a link or other access point.**

**A3:** No.

**4. Will the dataset be updated (e.g., to correct labeling errors, add new instances, delete instances)? If so, please describe how often, by whom, and how updates will be communicated to users (e.g., mailing list, GitHub)?**

**A4:** No. We have carefully three rounds of cross-checked the annotations to reduce the labeling errors. There may be very few labeling errors, which can be treated as noise.

**5. Will older versions of the dataset continue to be supported/hosted/maintained? If so, please describe how. If not, please describe how its obsolescence will be communicated to users.**

**A5:** N/A.

**6. If others want to extend/augment/build on/contribute to the dataset, is there a mechanism for them to do so? If so, please provide a description. Will these contributions be validated/verified? If so, please describe how. If not, why not? Is there a process for communicating/distributing these contributions to other users? If so, please provide a description.**

**A6:** N/A.

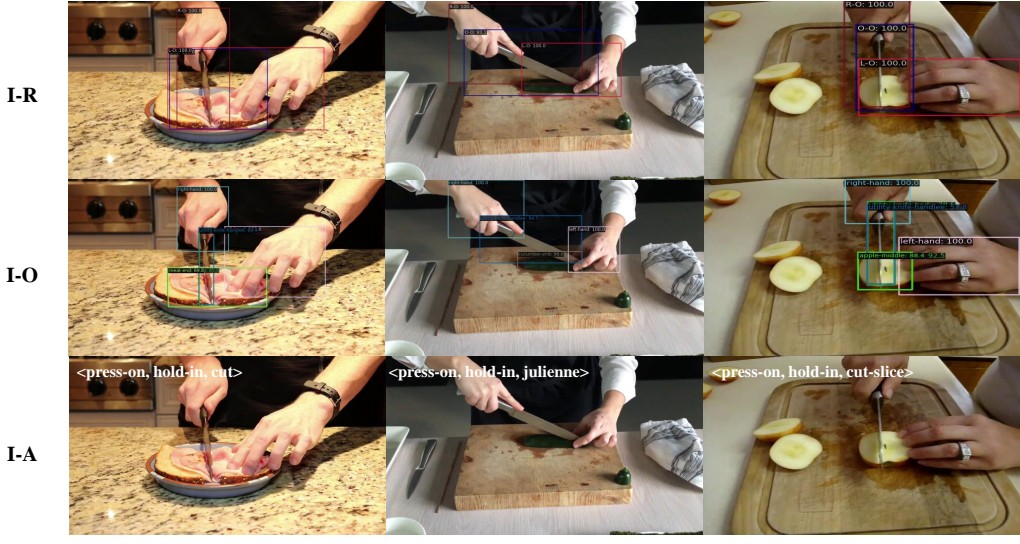

Figure 13: Some visual examples of three-hand interaction regions in our FHA-Kitchens. I-R: Interaction Region, I-O: Interaction Object, I-A: Interaction Region Action Verb

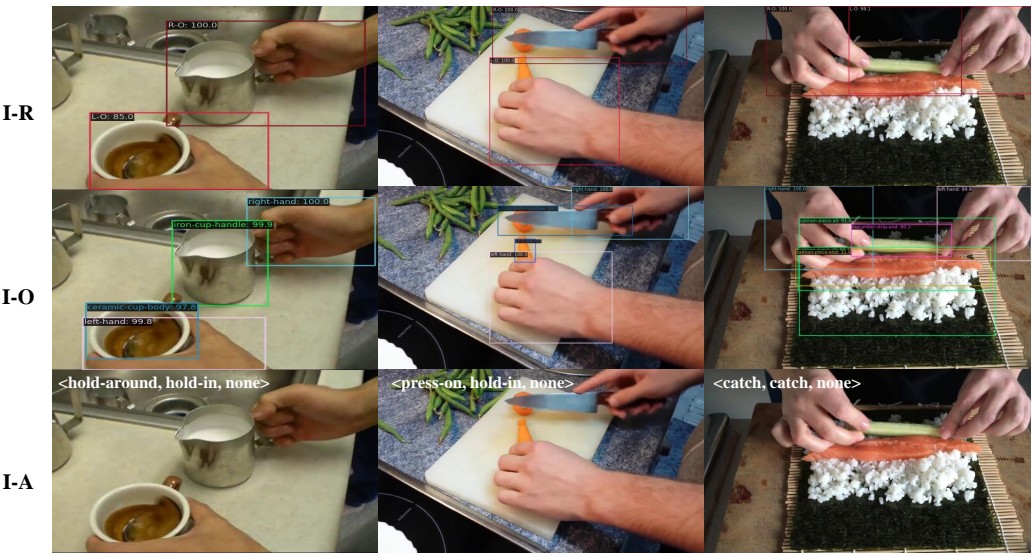

Figure 14: Some visual examples of two-hand interaction regions in our FHA-Kitchens. I-R: Interaction Region, I-O: Interaction Object, I-A: Interaction Region Action Verb

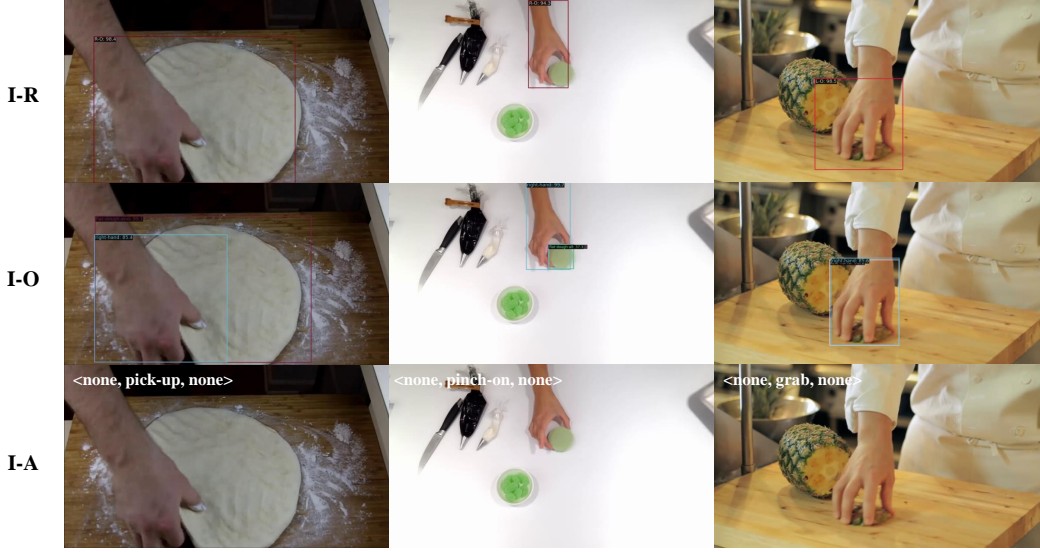

Figure 15: Some visual examples of one-hand interaction region in our FHA-Kitchens. I-R: Interaction Region, I-O: Interaction Object, I-A: Interaction Region Action Verb

Table 12: Vocabulary of fine-grained hand action verbs.

| ID | Verb | #Instance | ID | Verb | #Instance |
|---|---|---|---|---|---|
| 0 | hold-around | 1,593 | 66 | contain | 144 |
| 1 | hold-at | 788 | 67 | roll-on | 82 |
| 2 | fill-with | 265 | 68 | stick-to | 50 |
| 3 | pinch-on | 1,115 | 69 | touch-to | 12 |
| 4 | rub-around | 45 | 70 | smooth-out | 144 |
| 5 | hold-in | 20,520 | 71 | sprinkle-on | 203 |
| 6 | touch-on | 42 | 72 | squeeze-around | 189 |
| 7 | hold-with | 341 | 73 | press-down | 675 |
| 8 | press-on | 9,369 | 74 | cut-up | 100 |
| 9 | cut-out | 889 | 75 | shovel-up | 123 |
| 10 | fix-on | 2,037 | 76 | grab-out | 1 |
| 11 | peel-off | 1,413 | 77 | close | 51 |
| 12 | slice-along | 1,306 | 78 | rotate | 51 |
| 13 | grab | 2,531 | 79 | open | 72 |
| 14 | cut-half | 230 | 80 | open-down | 27 |
| 15 | take-up | 134 | 81 | ratate-around | 1 |
| 16 | pinch | 609 | 82 | hold-down | 41 |
| 17 | catch | 446 | 83 | cut-dice | 443 |
| 18 | put-down | 1,406 | 84 | dig-seeds | 124 |
| 19 | roll-up | 1,296 | 85 | chop | 346 |
| 20 | fix | 293 | 86 | push-forward | 8 |
| 21 | scrub-inside | 53 | 87 | cut-halves | 22 |
| 22 | lay-down | 111 | 88 | peel | 87 |
| 23 | hold-onto | 453 | 89 | push-ahead | 2 |
| 24 | pick-up | 526 | 90 | screw-on | 5 |
| 25 | cut-slice | 2,808 | 91 | sprinkle-into | 16 |
| 26 | take-out | 178 | 92 | scoop-up | 85 |
| 27 | turn-off | 22 | 93 | hold-along | 129 |
| 28 | cut-down | 1,040 | 94 | scrape-on | 331 |
| 29 | cut-off | 820 | 95 | stick-with | 13 |
| 30 | grab-up | 115 | 96 | cut-in | 275 |
| 31 | put-up | 56 | 97 | rub-on | 11 |
| 32 | break-apart | 221 | 98 | put-on | 2 |
| 33 | touch | 483 | 99 | push-off | 10 |
| 34 | cut-into-halves | 136 | 100 | place-on | 17 |
| 35 | bring-up | 140 | 101 | cut | 15 |
| 36 | pour-out | 832 | 102 | dip-in | 9 |
| 37 | pour-into | 395 | 103 | stretch-out | 31 |
| 38 | pour | 154 | 104 | flip | 8 |
| 39 | scrape | 70 | 105 | set-aside | 22 |
| 40 | rotate-around | 161 | 106 | julienne | 165 |
| 41 | screw-down | 19 | 107 | unroll | 270 |
| 42 | remove-out | 69 | 108 | adjust | 46 |
| 43 | hold-up | 664 | 109 | place-down | 262 |
| 44 | scoop-out | 76 | 110 | pile | 49 |
| 45 | open-up | 21 | 111 | pull | 137 |
| 46 | hold | 946 | 112 | attach-to | 18 |
| 47 | hold-on | 187 | 113 | grab-in | 15 |
| 48 | squeeze | 334 | 114 | knock-on | 29 |
| 49 | squeeze-out | 254 | 115 | press-against | 13 |
| 50 | mix-together | 187 | 116 | stir-in | 114 |
| 51 | spread-on | 342 | 117 | pull-up | 47 |
| 52 | twist-off | 20 | 118 | point-at | 25 |
| 53 | wrap-around | 204 | 119 | pull-out | 33 |
| 54 | break-off | 417 | 120 | scrape-down | 6 |
| 55 | grab-at | 621 | 121 | grab-onto | 558 |
| 56 | grab-on | 384 | 122 | hold-into | 88 |
| 57 | cut-through | 25 | 123 | hold-over | 67 |
| 58 | chop-dice | 63 | 124 | stir-into | 49 |
| 59 | sprinkle | 272 | 125 | press-onto | 1 |
| 60 | insert-into | 106 | 126 | roll | 68 |
| 61 | put-in | 30 | 127 | roll-out | 33 |
| 62 | dig-out | 23 | 128 | dip | 51 |
| 63 | cut-chunks | 91 | 129 | brush-onto | 54 |
| 64 | churn | 369 | 130 | flip-over | 83 |
| 65 | knead | 298 | | | |

Table 13: Vocabulary of fine-grained interaction object nouns.

| Super category | ID | Noun | #Instance | Super category | ID | Noun | #Instance |
|---|---|---|---|---|---|---|---|
| | 0 | basil-end | 201 | | 65 | apple-all | 22 |
| | 1 | beet-end | 143 | | 66 | apple-end | 34 |
| | 2 | beet-head | 136 | | 67 | apple-head | 253 |
| | 3 | beet-middle | 36 | | 68 | apple-middle | 301 |
| | 4 | bell-pepper-all | 2 | | 69 | avocado-end | 156 |
| | 5 | bell-pepper-end | 210 | | 70 | avocado-head | 12 |
| | 6 | bell-pepper-head | 67 | | 71 | avocado-left | 34 |
| | 7 | bell-pepper-middle | 139 | | 72 | avocado-middle | 174 |
| | 8 | broccoli-head | 38 | | 73 | avocado-right | 34 |
| | 9 | carrot-end | 1,625 | | 74 | block-watermelon-edge | 5 |
| | 10 | carrot-head | 205 | | 75 | green-melon-all | 4,074 |
| | 11 | carrot-middle | 638 | | 76 | green-melon-end | 68 |
| | 12 | chopped-vegetables-surface | 38 | | 77 | green-melon-middle | 68 |
| | 13 | courgette-end | 921 | | 78 | half-apple-head | 139 |
| | 14 | courgette-middle | 39 | | 79 | half-apple-middle | 47 |
| | 15 | cucumber-end | 288 | | 80 | half-pineapple-head | 6 |
| | 16 | cucumber-middle | 165 | | 81 | half-pineapple-middle | 99 |
| | 17 | cucumber-strip-all | 14 | | 82 | half-tomato-end | 1 |
| | 18 | cucumber-strip-end | 72 | | 83 | half-tomato-middle | 1 |
| | 19 | cucumber-strip-middle | 22 | | 84 | half-watermelon-edge | 79 |
| | 20 | garlic-middle | 240 | | 85 | half-watermelon-end | 17 |
| | 21 | garlic-end | 164 | | 86 | half-watermelon-head | 301 |
| | 22 | garlic-head | 46 | | 87 | half-watermelon-middle | 232 |
| | 23 | ginger-end | 248 | Fruits | 88 | lemon-end | 156 |
| | 24 | ginger-head | 169 | | 89 | lemon-middle | 108 |
| | 25 | ginger-middle | 70 | | 90 | melon-skin-all | 119 |
| | 26 | green-beans-end | 937 | | 91 | melon-skin-end | 576 |
| | 27 | green-pepper-dice | 1 | | 92 | melon-pulp-all | 28 |
| | 28 | green-pepper-end | 710 | | 93 | melon-pulp-end | 163 |
| | 29 | green-pepper-head | 142 | | 94 | melon-pulp-middle | 49 |
| | 30 | green-pepper-middle | 505 | | 95 | melon-slice-end | 200 |
| | 31 | half-bell-pepper-end | 116 | | 96 | orange-all | 22 |
| | 32 | half-bell-pepper-middle | 110 | | 97 | orange-end | 24 |
| | 33 | half-onion-all | 23 | | 98 | orange-head | 209 |
| | 34 | half-onion-head | 11 | | 99 | orange-middle | 547 |
| | 35 | half-onion-middle | 11 | | 100 | peeless-orange-middle | 93 |
| Vegetables&Plants | 36 | mushroom-middle | 15 | | 101 | piece-orange-edge | 276 |
| | 37 | nori-all | 506 | | 102 | pineapple-all | 26 |
| | 38 | nori-end | 262 | | 103 | pineapple-end | 476 |
| | 39 | onion-end | 78 | | 104 | pineapple-head | 959 |
| | 40 | onion-head | 28 | | 105 | pineapple-middle | 1,346 |
| | 41 | onion-middle | 49 | | 106 | slice-pineapple-end | 25 |
| | 42 | pepper-seeds-all | 4 | | 107 | slice-pineapple-middle | 63 |
| | 43 | piece-onion-middle | 38 | | 108 | watermelon-edge | 29 |
| | 44 | piece-tomato-end | 41 | | 109 | watermelon-end | 966 |
| | 45 | purple-cabbage-end | 77 | | 110 | watermelon-head | 155 |
| | 46 | purple-cabbage-head | 70 | | 111 | watermelon-middle | 631 |
| | 47 | purple-cabbage-middle | 23 | | 112 | boiled-egg-end | 226 |
| | 48 | red-pepper-all | 21 | | 113 | boiled-egg-head | 22 |
| | 49 | red-pepper-head | 25 | | 114 | boiled-egg-middle | 14 |
| | 50 | red-pepper-middle | 18 | | 115 | boiled-egg-shell | 88 |
| | 51 | small-tomato-head | 9 | | 116 | egg-all | 248 |
| | 52 | small-tomato-middle | 27 | | 117 | egg-head | 1 |
| | 53 | spinach-end | 4 | Dairy&Eggs | 118 | egg-middle | 26 |
| | 54 | spinach-head | 35 | | 119 | egg-liquid-all | 10 |
| | 55 | spinach-middle | 30 | | 120 | egg-shell-all | 86 |
| | 56 | spring-garlic-all | 18 | | 121 | egg-shell-edge | 34 |
| | 57 | spring-garlic-end | 53 | | 122 | milk-all | 594 |
| | 58 | spring-garlic-head | 23 | | 123 | yolk-all | 112 |
| | 59 | spring-garlic-middle | 52 | | 124 | chicken-dice | 8 |
| | 60 | sun-flower-seeds | 104 | | 125 | raw-chicken-dice | 69 |
| | 61 | tomato-cube | 22 | Meat&Fish | 126 | crab-shred | 328 |
| | 62 | tomato-end | 280 | | 127 | meat-end | 515 |
| | 63 | tomato-middle | 17 | | 128 | meat-head | 573 |
| | 64 | tomato-sliced-middle | 109 | | 129 | meat-middle | 142 |

Table 14: Vocabulary of fine-grained interaction object nouns.

| Super category | ID | Noun | #Instance | Super category | ID | Noun | #Instance |
|---|---|---|---|---|---|---|---|
| Meat&Fish | 130 | meat-piece-end | 442 | Containers | 194 | bottle-body | 52 |
| | 131 | meat-slice-all | 1 | | 195 | box-lid-bottom | 21 |
| | 132 | meat-slice-end | 202 | | 196 | bottle-cap | 8 |
| | 133 | piece-pepperoni-all | 87 | | 197 | bottle-cap-all | 106 |
| | 134 | piece-pepperoni-end | 158 | | 198 | bottle-cap-bottom | 19 |
| | 135 | salmon-piece-all | 36 | | 199 | can-cover-edg | 2 |
| | 136 | salmon-piece-end | 399 | | 200 | can-cover-edge | 106 |
| | 137 | salmon-piece-middle | 14 | | 201 | ceramic-cup-all | 68 |
| | 138 | salmon-slice-end | 44 | | 202 | ceramic-cup-body | 671 |
| Spices&Sauces | 139 | butter-all | 45 | | 203 | ceramic-cup-handle | 12 |
| | 140 | crumbles-cheese-all | 116 | | 204 | ceramic-lid-all | 25 |
| | 141 | cheese-all | 27 | | 205 | ceramic-lid-edge | 85 |
| | 142 | green-butter-all | 85 | | 206 | ceramic-teapot-handle | 132 |
| | 143 | mozzarella-all | 84 | | 207 | ceramic-teacup-body | 69 |
| | 144 | mozzarella-end | 12 | | 208 | ceramic-teacup-edge | 138 |
| | 145 | pizza-sauce-end | 193 | | 209 | cup-edge | 79 |
| | 146 | powder-all | 29 | | 210 | glass-bottle-edge | 40 |
| | 147 | sauce-all | 397 | | 211 | glass-bottle-top | 59 |
| | 148 | slice-cheese-end | 44 | | 212 | glass-cup-body | 1 |
| | 149 | tomato-sauce-all | 151 | | 213 | glass-cup-edge | 217 |
| | 150 | tomato-sauce-edge | 15 | | 214 | glass-cup-handle | 51 |
| | 151 | sauce-mixed | 70 | | 215 | glass-goblet-stem | 333 |
| Liquids | 152 | can-opener | 108 | | 216 | glastic-bottle-edge | 4 |
| | 153 | green-mixture-all | 248 | | 217 | glastic-bottle-top | 4 |
| | 154 | jam-all | 23 | | 218 | grass-bottle-top | 30 |
| | 155 | oil-all | 60 | | 219 | iron-basin-body | 29 |
| | 156 | olive-oil-all | 51 | | 220 | iron-basin-edge | 145 |
| Baked&Baking | 157 | baking-paper-edge | 99 | | 221 | iron-basin-middle | 115 |
| | 158 | baking-paper-top | 25 | | 222 | iron-cup-body | 1,005 |
| | 159 | baking-plate-edge | 54 | | 223 | iron-cup-handle | 325 |
| Cooked Food | 160 | piece-pizza-end | 125 | | 224 | iron-dipper-handle | 14 |
| | 161 | pizza-all | 63 | | 225 | plastic-basin-edge | 33 |
| | 162 | pizza-end | 27 | | 226 | plastic-bottle-bottom | 21 |
| | 163 | pizza-middle | 29 | | 227 | plastic-bottle-edge | 352 |
| | 164 | sandwich-edge | 32 | | 228 | plastic-bottle-top | 78 |
| | 165 | sandwich-end | 297 | | 229 | plastic-cup-body | 19 |
| | 166 | sandwich-head | 141 | | 230 | sauce-container-end | 4 |
| | 167 | sandwich-middle | 123 | | 231 | small-cup-edge | 195 |
| | 168 | sandwich-side | 85 | | 232 | small-plastic-bottle-edge | 154 |
| | 169 | sandwich-top | 27 | | 233 | small-plastic-bottle-end | 120 |
| | 170 | sandwich-all | 23 | | 234 | small-plastic-bottle-top | 167 |
| | 171 | sushi-roll-all | 151 | | 235 | teapot-lid-edge | 126 |
| | 172 | sushi-roll-end | 1,659 | | 236 | teapot-lid-handle | 267 |
| | 173 | sushi-roll-head | 233 | | 237 | wine-bottle-bottom | 75 |
| | 174 | sushi-roll-middle | 622 | | 238 | yogurt-box-bottom | 63 |
| Packaging | 175 | bamboo-mat-edge | 284 | | 239 | yogurt-box-edge | 62 |
| | 176 | bamboo-mat-end | 528 | | 240 | yogurt-box-handle | 2 |
| | 177 | bamboo-mat-head | 8 | | 241 | yogurt-box-top | 14 |
| | 178 | bamboo-mat-middle | 244 | | 242 | sauce-cup-all | 9 |
| | 179 | mozzarella-bag-end | 71 | Cutlery | 243 | bowl-bottom | 69 |
| | 180 | mozzarella-bag-middle | 24 | | 244 | bowl-edge | 198 |
| | 181 | onion-bag-end | 86 | | 245 | glass-bowl-all | 23 |
| | 182 | onion-bag-middle | 30 | | 246 | glass-bowl-body | 23 |
| | 183 | pepperoni-bag-end | 60 | | 247 | glass-bowl-bottom | 91 |
| | 184 | pepperoni-bag-middle | 16 | | 248 | glass-bowl-edge | 415 |
| | 185 | piping-bag-all | 251 | | 249 | glass-bowl-handle | 8 |
| | 186 | pizza-box-edge | 140 | | 250 | grass-bowl-edge | 13 |
| | 187 | tea-leaves-bag-body | 38 | | 251 | green-bowl-edge | 29 |
| | 188 | tea-leaves-bag-bottom | 42 | | 252 | small-bowl-edge | 16 |
| | 189 | tea-leaves-bag-top | 38 | | 253 | steel-bowl-edge | 153 |
| Containers | 190 | black-bottle-top | 42 | | 254 | steel-bowl-top | 13 |
| | 191 | bottle-all | 1 | | 255 | metal-bowl-edge | 470 |
| | 192 | bottle-edge | 19 | | 256 | plastic-bowl-all | 46 |
| | 193 | bottle-top | 30 | | | | |

Table 15: Vocabulary of fine-grained interaction object nouns.

| Super category | ID | Noun | #Instance | Super category | ID | Noun | #Instance |
|---|---|---|---|---|---|---|---|
| Cutlery | 257 | plastic-bowl-body | 10 | Kitchenware | 321 | shovel-body | 49 |
| | 258 | plastic-bowl-edge | 31 | | 322 | shovel-handle | 49 |
| | 259 | porcelain-bowl-edge | 39 | | 323 | sieve-spoon-body | 13 |
| | 260 | porcelain-bowl-middle | 2 | | 324 | sieve-spoon-handle | 12 |
| | 261 | porcelain-bowl-top | 17 | | 325 | small-iron-pot-handle | 90 |
| | 262 | fork-handle | 9 | | 326 | small-knife-body | 663 |
| | 263 | iron-spoon-body | 353 | | 327 | small-knife-handle | 706 |
| | 264 | iron-spoon-handle | 395 | | 328 | tea-strainer-body | 108 |
| | 265 | plastic-scoop-body | 48 | | 329 | tea-strainer-edge | 4 |
| | 266 | plastic-scoop-handle | 94 | | 330 | tea-strainer-handle | 224 |
| | 267 | plastic-spoon-body | 22 | | 331 | turnplate-corner | 15 |
| | 268 | plastic-spoon-handle | 22 | | 332 | turnplate-edge | 9 |
| | 269 | spoon-body | 692 | | 333 | utility-knife-body | 10,419 |
| | 270 | spoon-handle | 1,011 | | 334 | utility-knife-handle | 11,157 |
| | 271 | tablespoon-body | 327 | | 335 | carrot-peeler-body | 255 |
| | 272 | tablespoon-handle | 332 | | 336 | carrot-peeler-handle | 518 |
| | 273 | teaspoon-body | 51 | Appliances | 337 | grater-body | 70 |
| | 274 | teaspoon-handle | 191 | | 338 | grater-handle | 70 |
| | 275 | wooden-spoon-body | 116 | | 339 | oven-door-handle | 22 |
| | 276 | wooden-spoon-handle | 152 | Rice&Flour | 340 | ball-dough-end | 27 |
| | 277 | wooden-spatula-body | 25 | | 341 | ball-dough-head | 24 |
| | 278 | wooden-spatula-handle | 28 | | 342 | ball-dough-middle | 173 |
| | 279 | table-knife-handle | 34 | | 343 | dough-all | 413 |
| | 280 | tableware-handle | 11 | | 344 | dough-flour-all | 117 |
| Kitchenware | 281 | beater-end | 25 | | 345 | dough-flour-middle | 35 |
| | 282 | plate-edge | 335 | | 346 | flat-dough-all | 42 |
| | 283 | plate-end | 20 | | 347 | flat-dough-edge | 206 |
| | 284 | brush-body | 105 | | 348 | flat-dough-end | 963 |
| | 285 | brush-handle | 195 | | 349 | flat-dough-middle | 13 |
| | 286 | can-opener-edge | 83 | | 350 | flour-all | 35 |
| | 287 | can-opener-end | 25 | | 351 | green-dough-all | 123 |
| | 288 | ceramic-plate-all | 91 | | 352 | little-dough-al | 1 |
| | 289 | ceramic-plate-edge | 208 | | 353 | little-dough-all | 386 |
| | 290 | chopping-board-edge | 2 | | 354 | oval-dough-end | 170 |
| | 291 | cooking-spoon-body | 77 | | 355 | rice-all | 585 |
| | 292 | cooking-spoon-handle | 382 | | 356 | slice-bread-end | 92 |
| | 293 | food-mixer-handle | 372 | | 357 | bread-end | 510 |
| | 294 | food-mixer | 369 | | 358 | bread-head | 199 |
| | 295 | pizza-cutter-body | 29 | | 359 | bread-middle | 258 |
| | 296 | pizza-cutter-handle | 17 | Dessert | 360 | chocolate-cake-edge | 242 |
| | 297 | food-plate-center | 75 | | 361 | chocolate-cake-top | 111 |
| | 298 | food-plate-edge | 36 | | 362 | chocolate-all | 21 |
| | 299 | handle-all | 108 | | 363 | candies-all | 159 |
| | 300 | iron-plate-edge | 76 | | 364 | candy-all | 4 |
| | 301 | kitchen-knife-body | 552 | | 365 | candy-top | 51 |
| | 302 | kitchen-knife-handle | 1,215 | | 366 | chocolate-bar-middle | 87 |
| | 303 | knife-body | 26 | | 367 | chocolate-chips-all | 49 |
| | 304 | knife-handle | 27 | | 368 | chocolate-cream-all | 189 |
| | 305 | jam-knife-body | 50 | | 369 | chocolate-cream-top | 261 |
| | 306 | jam-knife-handle | 53 | | 370 | chocolate-donut-all | 35 |
| | 307 | metal-plate-edge | 47 | | 371 | cookie-all | 41 |
| | 308 | metal-spatula-body | 71 | | 372 | cookie-end | 57 |
| | 309 | metal-spatula-handle | 123 | | 373 | cookie-head | 1 |
| | 310 | pizza-spatula-body | 70 | | 374 | cookies-all | 49 |
| | 311 | pizza-spatula-handle | 99 | | 375 | cookie-top | 9 |
| | 312 | pizza-tray-edge | 122 | | 376 | cream-all | 139 |
| | 313 | plastic-spatula-body | 328 | Drink | 377 | tea-all | 14 |
| | 314 | plastic-spatula-handle | 531 | | 378 | tea-leaves-all | 219 |
| | 315 | rolling-pin-body | 198 | | 379 | whisk-head-top | 9 |
| | 316 | rolling-pin-handle | 174 | Uncategorised | 380 | hand-left | 23,305 |
| | 317 | rolling-pin-middle | 82 | | 381 | hand-right | 26,441 |
| | 318 | rolling-pin-miidle | 5 | | 382 | towel-all | 82 |
| | 319 | serrated-knife-body | 38 | | 383 | towel-edge | 38 |
| | 320 | serrated-knife-handle | 43 | | | | |

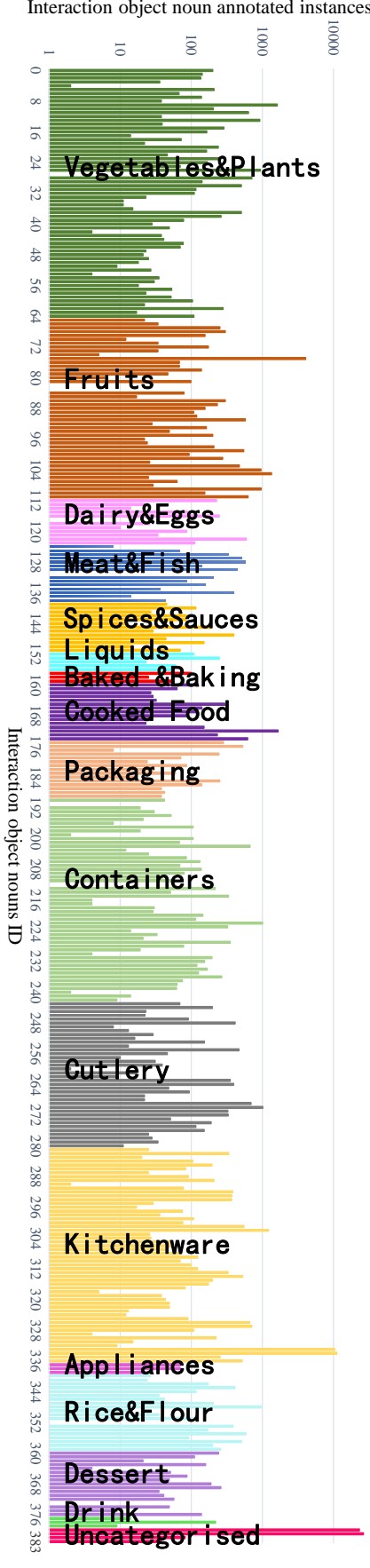

Figure 16: The distribution of instances per object noun category from 17 super-categories in the FHA-Kitchens dataset.

