# OpenReview forum: "FHA-Kitchens: A Novel Dataset for Fine-Grained Hand Action Recognition in Kitchen Scenes"
_ICLR.cc/2024/Conference — Submitted to ICLR 2024_

### Official Review · Reviewer_kr7t · 2023-10-18

**Soundness:** 3 good
**Presentation:** 3 good
**Contribution:** 2 fair
**Rating:** 5
**Confidence:** 3

**Summary:**

This paper proposes a dataset for fine-grained hand action recognition in kitchen scenes. It include three tasks, i.e., detection, recognition, and domain generalization.

**Strengths:**

1.Dataset contribution is meaningful to the community.
2.Human action understanding is worth studying.
3.This paper is easy to read.
4. 'Verb' categories are abundant.

**Weaknesses:**

1. There are also fine-grained human action datasets, such as fine-gym.
2. In this dataset, action categories are highly related with objects. Then object bias may appear, which means object will help the model understand actions.  I wonder about the task difficulty. Furthermore, it may cause researcher focus on objects rather than actions.
3. Following the point 2, can you just classify the 'verb' category?
4. No correponding baseline model is proposed.

**Questions:**

Refer to the weakness.

---

### Official Review · Reviewer_9rBA · 2023-10-31

**Soundness:** 3 good
**Presentation:** 2 fair
**Contribution:** 2 fair
**Rating:** 3
**Confidence:** 5

**Summary:**

The paper presents "FHA-Kitchens", a new dataset tailored for fine-grained hand action recognition within kitchen environments. This proposed dataset emphasizes human hand interactions in the kitchen, providing a more detailed perspective than traditional datasets that often focus on broader, full-body actions.

**Strengths:**

1. The FHA-Kitchens dataset is meticulously curated, with videos spanning diverse dish preparations, offering a holistic view of kitchen interactions.
2. The comprehensive benchmarking across three distinct paradigms provides valuable insights into the dataset's utility and the challenges inherent in hand action recognition.

**Weaknesses:**

1. While the dataset's focus is niche, the total number of videos (30) might be considered limited, especially when compared to other large-scale datasets.
2. The exclusive focus on kitchen environments might limit the dataset's applicability to broader hand action recognition scenarios outside of kitchen contexts.
3. The proposed dataset is quite similar to the EPIC-KITCHENS-100 dataset [1]. Upon comparison, it appears that the proposed dataset is of a smaller scale and offers fewer tracks than the EPIC-KITCHENS-100. I would appreciate further clarification on its unique contributions to the community.



[1] Damen D, Doughty H, Farinella G M, et al. Rescaling egocentric vision: Collection, pipeline and challenges for epic-kitchens-100[J]. International Journal of Computer Vision, 2022: 1-23.

**Questions:**

1. In the process of generating object segment information, could you elaborate on how SAM is utilized to produce the object mask? To my understanding, SAM typically outputs a class-agnostic object mask. How do you subsequently determine the class label?
2. The proposed dataset is quite similar to the EPIC-KITCHENS-100 dataset [1]. Upon comparison, it appears that the proposed dataset is of a smaller scale and offers fewer tracks than the EPIC-KITCHENS-100. I would appreciate further clarification on its unique contributions to the community.

---

### Official Review · Reviewer_hqWJ · 2023-10-31

**Soundness:** 3 good
**Presentation:** 3 good
**Contribution:** 2 fair
**Rating:** 5
**Confidence:** 4

**Summary:**

The paper proposes a video dataset for hand action recognition. Specifically, the videos are specific to the Kitchen environment. They provide labels for fine-grained action classes and bounding box annotations for hands. Based on such data and annotations, the paper benchmarks three tasks: hand interaction and object detection, fine-grained hand action recognition, and domain generalization for hand region detection.

**Strengths:**

- The paper is well-written and easy to follow.
- The proposed dataset is quality-controlled and has extensive annotations.
- The dataset could be helpful to the computer vision community to address hand interaction understanding.
- The annotations provided in the form of hand-action triplets are novel and fine-grained.
- The authors benchmark the dataset on three important tasks.

**Weaknesses:**

- In Figure 1, why is there no annotation for the chopping board in O-O interaction? Also, there is carrot & chopping board interaction. Should there be bounding box annotations to highlight this? While the authors mention that they want to perform fine-grained hand action recognition, it appears that many objects are not annotated.
- The number of frames in relatively low compared to modern hand datasets such as EPIC-Kitchens/ VISOR. There are only around 30K frames. Also, the dataset captures only eight dish categories.
- Do you provide contact annotations between hands and objects?
- Are there segmentation mask annotations for all the bounding boxes?
- There are important missing comparisons with existing relevant datasets such as EPIC-Kitchens-VISOR [1]. The VISOR benchmarks have high-quality annotations for hands and objects. The authors should compare their dataset against VISOR regarding annotations and cross-dataset evaluation. That is, there should be a comparison in evaluation performance of methods trained on FHA-Kitchen on VISOR, and vice-versa. This will help to understand the benefits of the proposed dataset.

[1] EPIC-KITCHENS VISOR Benchmark VIdeo Segmentations and Object Relations, NeurIPS 2022

**Questions:**

Please see the Weakness section

---

### Official Review · Reviewer_wEMj · 2023-11-02

**Soundness:** 2 fair
**Presentation:** 2 fair
**Contribution:** 2 fair
**Rating:** 3
**Confidence:** 4

**Summary:**

This paper proposes a fine-grained hand action annotation dataset that has 2,377 video clips and 30,047 images from 8 different types of dishes. The hand action is a triplet with a total number of 878 action triplets. Based on the proposed dataset, this paper benchmarked 3 tasks: (1) supervised learning for hand interaction region and object detection, (2) supervised learning for fine-grained hand action recognition, and (3) intra- and inter- class domain generalization for hand interaction region detection. The experiments show the performance of existing methods on the proposed dataset.

**Strengths:**

- This paper proposes a dataset with fine-grained hand action annotation. In the hand-object interaction part, it also includes the object-object interaction annotations which are useful for interaction learning tasks.
- It benchmarks on 3 tracks using existing methods.

**Weaknesses:**

- There is no method to sufficiently use the annotations collected in this paper, e.g. the 9-dim action labels.
- There is no method in the paper. In the experiments section, it runs exiting methods and ablates the backbones.
- The whole paper is about a dataset and three benchmarks thus it fits more to a dataset and benchmark track.

**Questions:**

- Table 1: This paper focuses on annotating hand action. There are many hand datasets and newer datasets that need to be incorporated into Table 1 for a comprehensive and fair comparison. For example,
    - Ego4D: Around the World in 3000 Hours of Egocentric Video. Grauman et al.
    - EPIC-KITCHENS VISOR Benchmark: VIdeo Segmentations and Object Relations. Darkhalil et al.
    - Understanding Human Hands in Contact at Internet Scale. Shan et al.
- In Table 1, there are more statistics about the dataset, for example, the number of hands, and objects. These are basic numbers of annotations and thus authors should show these numbers.
- In 3.1, it mentions that the authors collected 30 videos to make a balanced dataset. It is not clear how the 30 videos are collected, from YouTube or captured by the authors.
- In 3.2 Bounding Box Annotation of Hand Action: the same object, such as the knife in R-O and O-O in Figure 1 are annotated twice separately. The overlapping object bbox here is for the same object, the location annotation should be the same to maintain consistency of annotations. Why not only annotate the same object once?
- The paper emphasizes the contribution of annotating a high dimension of action label which has 9 dimensions. However, there is no experiments to use the 9-dim annotation or to support that these annotations are more helpful than existing ones.
- In the experiment section, Results on the SL-AR Track, the author mentioned that “However, all the models achieve much worse accuracy on our FHA-Kitchens than the coarse-grained Kinetics 400, and unsatisfactory performance even using the large models.” There is no results Kinetics evaluation set for comparison given that Table 4 is on the FHA Kitchens val set.

---

### Meta-Review · Area_Chair_mCy2 · 2023-12-09

**Metareview:**

The reviewers appreciate the efforts in collecting the dataset but question about the distinctions with respect to existing kitchen and hand datasets.  No rebuttal given.  All four reviewers recommend reject and the AC concurs.

**Justification For Why Not Higher Score:**

Contribution not clear with respect to existing datasets.  no rebuttal.

**Justification For Why Not Lower Score:**

N/A

---

### Decision · Program_Chairs · 2024-01-16

Reject